# The p38 MAP kinase pathway modulates the hypoxia response and glutamate receptor trafficking in aging neurons

Eun Chan Park[1,2], Christopher Rongo[1,2]*

[1]The Waksman Institute, Rutgers The State University of New Jersey, New Jersey, United States; [2]Department of Genetics, Rutgers The State University of New Jersey, New Jersey, United States

**Abstract** Neurons are sensitive to low oxygen (hypoxia) and employ a conserved pathway to combat its effects. Here, we show that p38 MAP Kinase (MAPK) modulates this hypoxia response pathway in *C. elegans*. Mutants lacking p38 MAPK components *pmk-1* or *sek-1* resemble mutants lacking the hypoxia response component and prolyl hydroxylase *egl-9*, with impaired subcellular localization of Mint orthologue LIN-10, internalization of glutamate receptor GLR-1, and depression of GLR-1-mediated behaviors. Loss of p38 MAPK impairs EGL-9 protein localization in neurons and activates the hypoxia-inducible transcription factor HIF-1, suggesting that p38 MAPK inhibits the hypoxia response pathway through EGL-9. As animals age, p38 MAPK levels decrease, resulting in GLR-1 internalization; this age-dependent downregulation can be prevented through either p38 MAPK overexpression or removal of CDK-5, an antagonizing kinase. Our findings demonstrate that p38 MAPK inhibits the hypoxia response pathway and determines how aging neurons respond to hypoxia through a novel mechanism.

**\*For correspondence:** rongo@waksman.rutgers.edu

**Competing interests:** The authors declare that no competing interests exist.

## Introduction

Whereas the brain comprises about 2% of human body weight, it consumes about 20% of oxygen intake, highlighting that neurons require robust aerobic energy production (*Clarke and Sokoloff, 1999*). Neurons do not have plentiful glycolytic reserves, yet must expend tremendous amounts of ATP to maintain their membrane potential. Failure to do so, as during the low oxygen conditions (hypoxia) that occur during ischemic stroke, can lead to the collapse of the membrane potential, large scale glutamate neurotransmitter release, and overactivation of glutamate receptors (*Takahashi et al., 2002*; *Lees, 2000*; *Xue et al., 1994*; *Li and Buchan, 1993*; *Gill, 1994*; *Sheardown et al., 1993*; *Nellgard and Wieloch, 1992*; *Buchan et al., 1991*). An understanding of how neurons respond to hypoxic stress is important for the development of new therapies to prevent and treat damage caused by ischemic stroke or traumatic injury.

Multicellular organisms respond to hypoxic stress by activating a conserved hypoxia response pathway, which includes prolyl hydroxylase enzymes as oxygen sensors and hypoxia-inducible factor alpha (HIFα) transcription factors as effectors. When oxygen levels are sufficiently high (i.e., normoxia), the EGL-9/PHD family of prolyl hydroxylases employs molecular oxygen, 2-oxoglutarate, and iron to hydroxylate specific proline side chains on the HIFα transcription factors, resulting in the inactivation of these transcription factors by ubiquitin-mediated protein degradation (*Fong and Takeda, 2008*; *Aragones et al., 2009*; *Wong et al., 2013*; *Majmundar et al., 2010*). When oxygen levels are not sufficiently high (i.e., hypoxia), the EGL-9/PHD prolyl hydroxylases become inactive, resulting in HIFα protein stabilization and thus a transcriptional change in gene expression (*Semenza, 2009*; *Fandrey and Gassmann, 2009*). Depending on the specific cells and tissues undergoing stress, HIFα

**eLife digest** The brain accounts for 2% of our body weight, but consumes about 20% of our oxygen intake. This oxygen gluttony is due to the tremendous appetite of brain cells for energy, which neurons satisfy through oxygen-dependent (aerobic) metabolism. As a result, the loss of oxygen to the brain during a stroke, heart attack, or due to another medical condition can be very damaging to cells in the brain.

Human and other animal cells use a communication system called the hypoxia response pathway to sense oxygen and trigger a protective response when oxygen is low. This pathway includes an enzyme called prolyl hydroxylase, which senses oxygen and modifies another protein in the pathway that regulates the production of enzymes involved in metabolism. This alters the balance of enzymes involved in aerobic and oxygen-independent (anaerobic) metabolism in the cell. However, it is not clear how the activity of the prolyl hydroxylase is regulated.

Much of our knowledge about the hypoxia response pathway has been gained from studies using a small worm called *C. elegans*. This worm uses the pathway to cope with hypoxia in the harsh environment of the soil. Mutant worms that lack the prolyl hydroxylase have several abnormalities including higher levels of anaerobic metabolism even in the presence of oxygen, and defects in the connections between neurons.

Park and Rongo used *C. elegans* to study the pathway in more detail. The experiments show that another enzyme called p38 MAPK activates the prolyl hydroxylase. Mutant worms that lack this enzyme have similar abnormalities in the hypoxia response pathway as animals that lack the prolyl hydroxylase. In normal worms, decreasing levels of p38 MAPK as the animals grow older contribute to the decline in the nervous system. The p38 MAPK enzyme appears to work by regulating the activity of the prolyl hydroxylase and its location inside neurons.

These findings provide a new target for the development of drugs that may help to protect us from tissue damage caused by hypoxia. Future challenges are to find out what activates p38 MAPK, and how it influences the location of prolyl hydroxylase in neurons.

can mediate adaptive responses to hypoxia that include increased erythropoiesis, increased angiogenesis, and reprogramming of metabolism away from oxidative phosphorylation and towards glycolysis and anaerobic fermentation. In addition to its role in response to acute hypoxic stress, the hypoxia response pathway also helps to maintain stem cell niches and tumor growth and metastasis in cancer (*Amelio and Melino, 2015*; *Gupta et al., 2014*; *Ito and Suda, 2014*). Whereas the major target of oxygen regulation through EGL-9/PHD proteins is HIFα, several studies have shown that EGL-9/PHD oxygen sensors regulate additional proteins as part of the overall hypoxia response (*Lee et al., 2005*; *Koditz et al., 2007*; *Fu et al., 2007*; *Fu and Taubman, 2010*; *Cummins et al., 2006*; *Park et al., 2012*; *Lee et al., 2015*). Thus, while the core pathway of the hypoxia response is well established, it is less clear how the pathway uses alternative effectors and modulators in different tissues and contexts so as to tailor the specific physiological response to stress.

Mammalian neurons are particularly sensitive to hypoxia, making in vivo studies challenging. The genetically tractable and hypoxia tolerant model organism *C. elegans* has allowed investigators to study how the hypoxia response pathway functions in multiple tissue types, developmental stages, and aging (*Rodriguez et al., 2013*; *Leiser and Kaeberlein, 2010*; *Powell-Coffman, 2010*). *C. elegans* possess a single prolyl hydroxylase, called EGL-9, and a single HIFα, called HIF-1. These two proteins are expressed in essentially all tissues in *C. elegans*, where they mediate the primary response that allows nematodes to survive when they encounter hypoxic niches within their natural environment of the soil. HIF-1 also has a complex role in regulating aging and protein homeostasis in *C. elegans* (*Rodriguez et al., 2013*; *Fawcett et al., 2015*).

Hypoxia modulates a specific nematode behavior through EGL-9 but independent of HIF-1 (*Park et al., 2012*). Nematodes navigate their environment using a biased random walk comprised of long runs of forward locomotion and spontaneous reversals of locomotion followed by changes in direction (*Cohen and Sanders, 2014*; *Gray et al., 2005*). The frequency of spontaneous reversals is determined by the activity of AMPA-type glutamate receptors (AMPARs) located in a small number

of command interneurons (*Schaefer and Rongo, 2006*; *Mellem et al., 2002*; *Zheng et al., 1999*). *C. elegans* avoids zones of hypoxia using a combination of sensory neuron-mediated aerotaxis and command interneuron-mediated spontaneous reversals (*Chang et al., 2006*; *Cheung et al., 2005*; *Park et al., 2012*). In a normoxic environment, *C. elegans* exhibits a relatively high frequency of spontaneous reversals, resulting in a bias towards local foraging behaviors. When exposed to hypoxia for long periods, *C. elegans* exhibits a depressed frequency of spontaneous reversals, resulting in a bias towards roaming behavior that allows the animal to potentially exit the hypoxic environment. The local foraging behavior in normoxic environments requires EGL-9, as *egl-9* mutants exhibit decreased reversals similar to those observed under hypoxia (*Park et al., 2012*). Surprisingly, this behavioral phenotype does not require HIF-1.

Hypoxia and EGL-9 regulate *C. elegans* reversal behavior by regulating the membrane trafficking of the AMPAR subunit GLR-1. GLR-1-containing AMPARs act in the command interneurons to receive synaptic input and direct overall locomotory reversal behavior (*Hart et al., 1995*; *Mellem et al., 2002*; *Maricq et al., 1995*; *Chang and Rongo, 2005*). Mutants that lack GLR-1 have a depressed frequency of spontaneous reversals. The synaptic localization of GLR-1 can be detected in vivo using a functional GLR-1::GFP chimeric protein, and mutants that fail to localize GLR-1 to synapses also have a depressed frequency of reversals (*Burbea et al., 2002*; *Glodowski et al., 2005*; *Schaefer and Rongo, 2006*; *Shim et al., 2004*; *Zheng et al., 1999*; *Rongo et al., 1998*). Wild-type animals exposed to hypoxia (or *egl-9* mutants under normoxia) accumulate GLR-1 receptors in internal endosomal compartments (*Park et al., 2012*). Under normoxia, oxygen promotes the interaction and endosomal recruitment of EGL-9 with LIN-10, an ortholog of the Mint/X11 scaffolding molecules, and LIN-10 in turn promotes GLR-1 recycling to the plasma membrane (*Whitfield et al., 1999*; *Glodowski et al., 2005*; *Park et al., 2009*; *2012*). Under hypoxia, EGL-9 releases LIN-10, allowing LIN-10 to be phosphorylated by the CDK-5 kinase. Phosphorylated LIN-10 is then released from endosomes, resulting in diminished GLR-1 recycling, depletion of synaptic GLR-1 by endocytosis without accompanying recycling, and decreased GLR-1-mediated reversal behavior (*Park et al., 2012*; *Juo et al., 2007*).

EGL-9 and oxygen regulate GLR-1 recycling through a novel HIF-1-independent mechanism, suggesting that different tissues can employ parts of the hypoxia response pathway for specialized functions, and that additional modulators and mediators of the pathway remain to be discovered. Here, we show that signaling through the kinases SEK-1 (p38 MAPKK) and PMK-1 (p38 MAPK) regulate GLR-1 recycling and GLR-1-mediated reversal behavior by modulating the hypoxia response pathway. Loss of function mutations in either *pmk-1 or sek-1* mimic the effects of hypoxia on GLR-1 trafficking and behavioral output. Wild-type SEK-1 and PMK-1 promote the endosomal localization of EGL-9 and LIN-10 in neurons under normoxia, and the effect of *sek-1* or *pmk-1* mutations on EGL-9/LIN-10 co-localization and GLR-1 recycling requires the activity of the CDK-5 kinase. Wild-type SEK-1 and PMK-1 also regulate HIF-1 throughout the organism. Older animals show reduced levels of activated PMK-1, GLR-1 internalization, and decreased GLR-1-mediated behaviors. The reduction of functional GLR-1 in older animals can be prevented through either the overexpression of PMK-1 or the removal of CDK-5. Our findings demonstrate that p38 MAPK is a modulator of the hypoxia response pathway through EGL-9, and that this novel mechanism helps determine how aging neurons respond to hypoxia.

## Results

### Signaling through p38 MAPK regulates GLR-1 AMPAR localization

We previously showed that the hypoxia response pathway regulates GLR-1 recycling and function (*Park et al., 2012*), and we therefore reasoned that other signaling pathways that respond to oxidative stress conditions might also contribute to GLR-1 regulation. One such signaling molecule is the p38 MAPK ortholog PMK-1, which is involved in oxidative stress response and innate immunity (*Berman et al., 2001*; *Kim et al., 2002*; *Inoue et al., 2005*). To determine if this p38 MAPK regulates GLR-1 trafficking, we obtained a viable mutant strain homozygous for a complete loss of function (deletion) allele in *pmk-1* (*Mizuno et al., 2004*). We introduced a transgene, *nuIs25*, which expresses full length, functional GLR-1 receptors tagged with GFP (GLR-1::GFP), into *pmk-1* mutants. In wild-type nematodes, GLR-1::GFP is localized to discrete puncta (mean diameter of 0.48 microns, SEM of 0.01 microns, 95% of puncta are between 0.32 and 0.73 microns in diameter) along

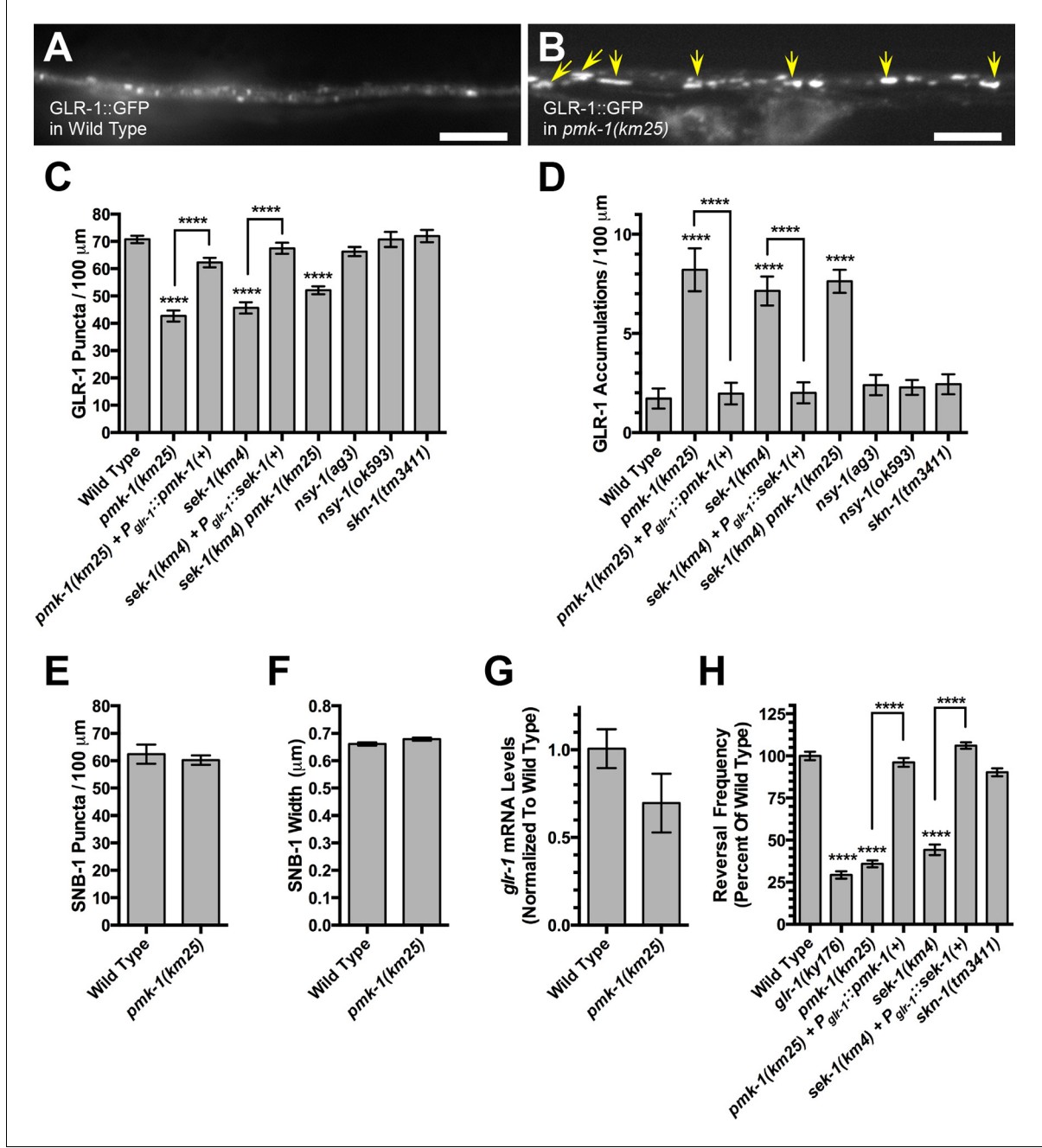

**Figure 1.** Signaling through PMK-1 p38 MAPK regulates GLR-1 AMPAR trafficking. GLR-1::GFP fluorescence in (A) wild-type animals and (B) *pmk-1 (km25)* mutants. GLR-1 is localized to elongated accumulations (indicated by yellow arrows). Bar: 5 μm. Average GLR-1::GFP number is quantified as (C, I) puncta or (D, J) accumulations per length of ventral cord dendrites. Average SNB-1::GFP puncta are quantified based on (E) number per length of ventral cord and (F) puncta width. (G) Relative *glr-1* mRNA levels quantified by qRT-PCR and normalized to the mean value for wild type. (H) Spontaneous reversal frequency (number of reversals measured over a 5-min period) represented as a percentage of the mean value for wild type. Graph bar columns labeled with asterisks indicate statistical difference by ANOVA followed by Dunnett's multiple comparison to wild type (****p<0.0001, ***p<0.001, **p<0.01, *p<0.05). Lines connecting specific columns indicate pairwise comparisons using the Holm-Šídák test. Error bars indicate SEM. N = 13–47 animals per genotype.

The following figure supplement is available for figure 1:

**Figure supplement 1.** Additional related factors that do not Alter GLR-1 localization In *C. elegans* neurons.

the ventral cord dendrites of interneurons (*Figure 1A*), with 85% of such puncta colocalized with synaptic markers (*Rongo et al., 1998*; *Burbea et al., 2002*). We found that *pmk-1* mutants accumulated GLR-1::GFP in elongated structures (mean length of 2.30 microns, SEM of 0.10 microns, 95% of these accumulations are between 1.32 and 3.49 microns in length) along the ventral cord (*Figure 1B*), similar to the GLR-1::GFP accumulations in elongated endosomes observed in mutants for membrane recycling factors (*Shi et al., 2010*; *Glodowski et al., 2007*; *Kramer et al., 2010*; *Park et al., 2009*; *Rongo et al., 1998*). GLR-1 puncta and GLR-1 accumulations are distinct enough in shape and size to allow easy quantification of their respective numbers along the ventral cord dendrites (*Figure 1C,D*). We found a sizeable decrease in the number of GLR-1 puncta (*Figure 1C*) and a five-fold increase in the number of GLR-1 elongated accumulations (*Figure 1D*) in *pmk-1* mutants relative to wild type. Expression of a wild-type *pmk-1* cDNA from the *glr-1* promoter, which drives expression specifically in the command interneurons, was sufficient to rescue *pmk-1* mutants, indicating a cell autonomous requirement for PMK-1 function (*Figure 1C,D*). The observed changes in GLR-1 were unlikely to be due to general defects in synapse formation or overall cell polarity, as the localization of a synaptobrevin-GFP reporter (SNB-1::GFP), which decorates synaptic vesicles at interneuron presynaptic elements when expressed from a transgene, did not change in *pmk-1* mutants relative to wild type (*Figure 1E,F*). In addition, we detected similar levels of *glr-1* mRNA in wild-type animals and *pmk-1* mutants, indicating that PMK-1 regulates GLR-1 in a posttranscriptional fashion (*Figure 1G*).

PMK-1 is part of a p38 MAPK pathway that responds to bacterial infection and promotes innate immunity via the MAPKKK NSY-1 and the MAPKK SEK-1 (*Mizuno et al., 2004*). To determine if these signaling molecules are required to regulate GLR-1 trafficking, we examined GLR-1::GFP in loss of function mutations for both genes. A loss of function allele for *sek-1* resulted in a similar GLR-1 localization phenotype to that observed in *pmk-1* mutants (*Figure 1C,D*). Expression of a wild-type *sek-1* cDNA from the *glr-1* promoter rescued *sek-1* mutants, indicating a cell autonomous requirement for SEK-1 function (*Figure 1C,D*). Double mutants for *sek-1* and *pmk-1* showed a similar phenotype compared to either single mutant (*Figure 1C,D*), suggesting that these mutations do not yield an additive phenotype and are thus likely acting in the same pathway. By contrast, two loss of function, putative null alleles in *nsy-1* (one a nonsense mutation and the other an insertion resulting in a frameshift), did not result in abnormal GLR-1 localization (*Figure 1C,D*). Thus, whereas SEK-1 appears to be the MAPKK for PMK-1 to regulate GLR-1 localization, NSY-1 is unlikely to be the p38 MAPKKK. We examined mutants for several additional MAPKKK genes in *C. elegans*, including TIR-1, MLK-1, and KIN-18; however, we did not observe changes in GLR-1::GFP localization (*Figure 1—figure supplement 1*). Mutants for MAPKKK DLK-1 also do not show the same GLR-1::GFP localization defect observed in *pmk-1* mutants (*Park et al., 2009*). Our results suggest that the upstream MAPKKK for this function of p38 MAPK signaling is likely to be a noncanonical kinase relative to traditional MAPK signaling.

The *C. elegans* genome contains an additional p38 MAPK, called PMK-2, that functions redundantly with PMK-1 in the nervous system to regulate behavioral responses to pathogenic bacteria (*Pagano et al., 2015*). We examined GLR-1:GFP in *pmk-2(qd284)* mutants, which contain a deletion and frameshift at the beginning of the ORF, making this mutant allele a likely null; however, we did not observe a difference in GLR-1 puncta or accumulations compared to wild type (*Figure 1—figure supplement 1*). The gain of function mutation *pmk-2(qd171qd279)*, which can suppress other nervous system defects of *pmk-1(km25)* mutations (*Pagano et al., 2015*), did not suppress the effects of *pmk-1(km25)* mutations on GLR-1 localization (*Figure 1—figure supplement 1*). Two independent *pmk-2* loss of function mutations – *qd279* and *qd280* – did not enhance the GLR-1 localization phenotype caused by *pmk-1(km25)* mutations (*Figure 1—figure supplement 1*), although *qd280* showed a mild suppression of the depressed GLR-1 puncta number phenotype caused by the *pmk-1 (km25)* mutation. Taken together, our results indicate the PMK-2 does not regulate GLR-1 localization either by itself or redundantly with PMK-1.

PMK-1 signaling promotes innate immunity and the oxidative stress response by phosphorylating and activating the Nrf2 transcription factor ortholog SKN-1 (*Hoeven et al., 2011*; *Papp et al., 2012*). We therefore examined GLR-1::GFP in *skn-1* deletion allele homozygotes; however, we did not detect a difference in GLR-1::GFP localization relative to wild type (*Figure 1C,D*). PMK-1 also regulates the transcription factor ATF-7 (*Shivers et al., 2010*). We examined *atf-7* single mutants and *pmk-1 atf-7* double mutants; however, we did not observe a change in GLR-1::GFP localization

(*Figure 1—figure supplement 1*). These findings suggest that PMK-1 does not regulate GLR-1 through its most well established transcriptional outputs.

We also examined GLR-1::GFP localization in wild-type animals raised on the pathogenic bacteria *Pseudomonas aeruginosa* (strain PA14), which is known to promote an innate immune response by activating the PMK-1 pathway (*Papp et al., 2012*; *Shivers et al., 2010*) however, we did not observe any significant changes relative to wild type (*Figure 1—figure supplement 1*). These results indicate that the MAPKK SEK-1 and the MAPK PMK-1 act in a distinct and novel p38 MAPK signaling pathway to regulate GLR-1 subcellular localization.

## Signaling by p38 MAPK promotes GLR-1 AMPAR function and recycling from endosomes

The elongated accumulations of GLR-1 in *sek-1* and *pmk-1* mutants are similar to those observed in mutants in which GLR-1 recycling is impaired, suggesting that they might represent internalized receptors. Internalization of GLR-1 AMPARs results in diminished interneuron synaptic function. We examined functional synaptic GLR-1 through a standardized measurement: the frequency of spontaneous reversals of locomotion in the brief absence of food (*Schaefer and Rongo, 2006*; *Mellem et al., 2002*; *Zheng et al., 1999*). Wild-type animals exhibited a robust frequency of reversals, whereas *glr-1* null mutants showed a depressed reversal frequency (*Figure 1H*). Similar to *glr-1* null mutants, null mutants for *sek-1* and *pmk-1* showed a reduced frequency of reversals, whereas a null mutant for *skn-1* showed a reversal frequency that was similar to that of wild type (*Figure 1H*). Expression of either a wild-type *sek-1* or *pmk-1* cDNA from the *glr-1* promoter was sufficient to rescue the reversal phenotype of the corresponding mutation (*Figure 1H*). Thus, SEK-1 and PMK-1 are required to promote GLR-1 function.

If the elongated accumulations containing GLR-1::GFP in *pmk-1* and *sek-1* mutants represent AMPARs trapped in endosomes following endocytosis, then a reduction in GLR-1 endocytosis should suppress the accumulation of GLR-1 in these structures. We previously showed that expression of a dominant negative RAB-5, which contains a mutation that mimics the GDP-bound state of this GTPase, reduces GLR-1 endocytosis and suppresses internal accumulation of GLR-1 in membrane recycling mutants (*Park et al., 2009*; *Bucci et al., 1992*; *Li et al., 1994*). We introduced a transgene that expresses RAB-5(GDP) from the *glr-1* promoter into either *sek-1* or *pmk-1* mutants. For both mutants, expression of RAB-5(GDP) restored the GLR-1 synaptic puncta number to wild-type levels (*Figure 2A–D*) and suppressed the accumulation of GLR-1::GFP in elongated accumulations (*Figure 2C,E*), consistent with such accumulations being post-endocytic.

While the small size of *C. elegans* neurites has precluded an analysis of dendritic endosomes in the command interneurons, accumulation of GLR-1 in endosomes can be directly visualized in *C. elegans* neuron soma by examining GLR-1::GFP colocalization with an mRFP-tagged syntaxin (SYX-7) that resides at early endosomes (*Chun et al., 2008*; *Park et al., 2009*). We co-expressed GLR-1::GFP with mRFP::SYX-7 using the *glr-1* promoter and collected single confocal optical sections of neuron cell bodies that express both chimeric proteins. We then quantified co-localization by measuring the fraction of pixels that contain fluorescent from both proteins when such fluorescent was above baseline (*Figure 2F–Q*). Approximately 30% of GLR-1::GFP is co-localized with mRFP::SYX-7 in wild-type animals (*Figure 2F-I,R*). By contrast, there is approximately a 50% increase in the portion of GLR-1::GFP that is co-localized with mRFP::SYX-7 in *sek-1* (*Figure 2J-M,R*) and *pmk-1* (*Figure 2N-Q,R*) mutants, indicating that GLR-1 accumulates at SYX-7-decorated endosomes in these mutants, at least in soma. Taken together, our results indicate that in the absence of SEK-1/PMK-1 signaling, GLR-1 receptors accumulate in elongated, internal, and post-endocytic compartments in the command interneurons, resulting in diminished GLR-1 function and behavioral output.

## Loss of p38 MAPK signaling occludes any additional effects of hypoxia on GLR-1 localization

The elongated structures containing GLR-1::GFP observed in *sek-1* and *pmk-1* mutants resemble similar structures observed in wild-type nematodes exposed to hypoxia, as well as *egl-9* mutants under normoxia (*Park et al., 2012*). Given the role of PMK-1 in stress response, we reasoned that PMK-1 signaling might regulate GLR-1 recycling through EGL-9 and the hypoxia response pathway. To explore this possibility, we examined GLR-1::GFP in wild-type and *pmk-1* mutant animals

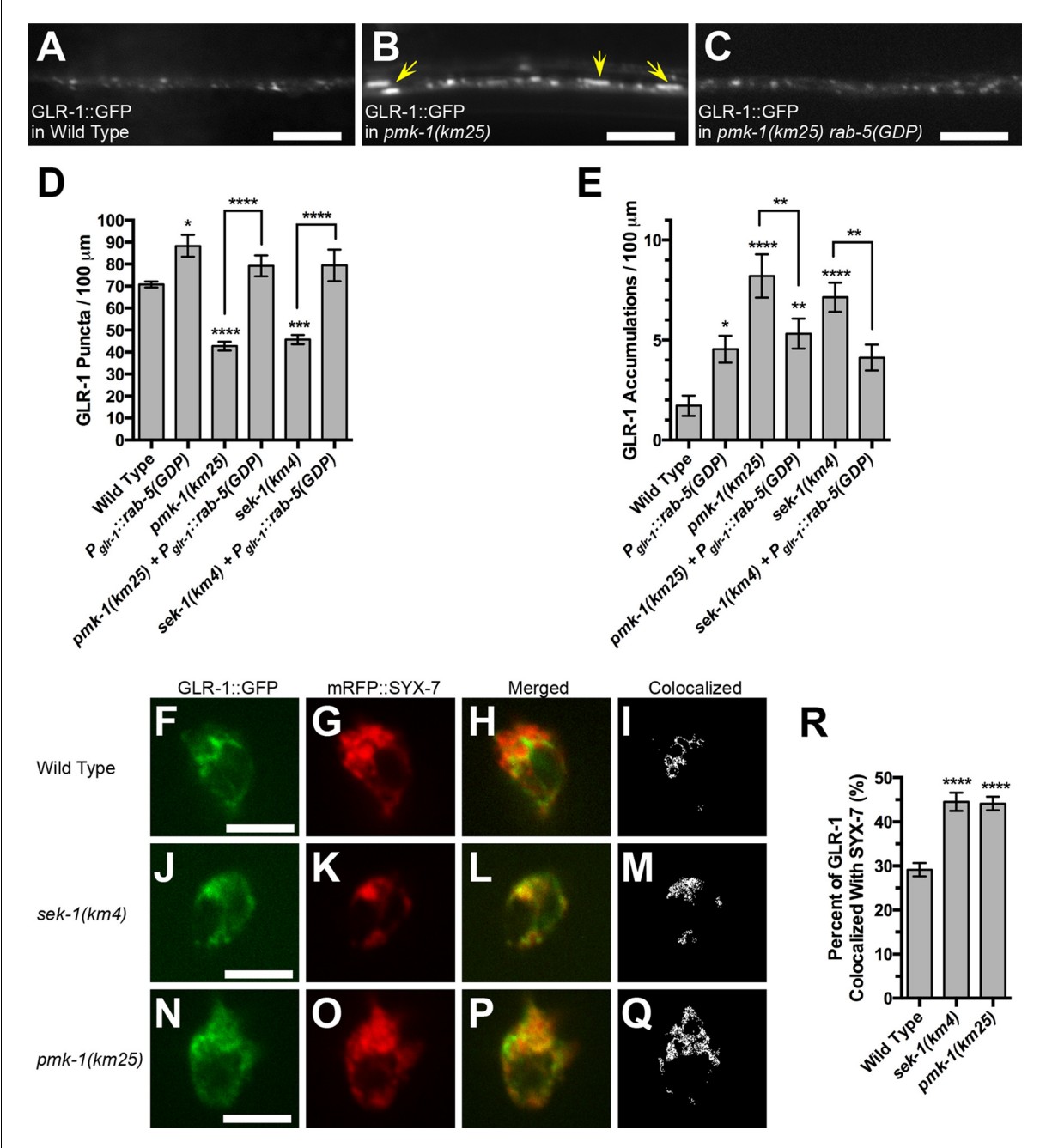

**Figure 2.** The p38 MAPK pathway promotes GLR-1 AMPAR function and recycling from endosomes. GLR-1::GFP fluorescence in (**A**) wild-type animals, (**B**) *pmk-1(km25)* mutants, and (**C**) *pmk-1(km25)* mutants containing a transgene that expresses dominant negative RAB-5 with a GDP-locked mutation. Yellow arrows indicate elongated accumulations. Bar: 5 μm. Average GLR-1::GFP number is quantified as (**D**) puncta or (**E**) accumulations per length of ventral cord dendrites. (**F, J, N**) GLR-1::GFP and (**G, K, O**) mRFP::SYX-7 fluorescence observed in the PVC neuron cell body of (**F, G, H, I**) wild type, (**J, K, L, M**) *sek-1* mutants, and (**N, O, P, Q**) *pmk-1* mutants. (**H, L, P**) Merged image of the red and green channels. (**I, M, Q**) Binary image with white indicating pixels with significant signal (colocalization) in both channels. (**R**) Fraction of GLR-1::GFP-labeled pixels that overlap with mRFP:SYX-7-labeled pixels. Graph bar columns labeled with asterisks indicate statistical difference by ANOVA followed by Dunnett's multiple comparison to wild type (****p<0.0001, ***p<0.001, **p<0.01, *p<0.05). Lines connecting specific columns indicate pairwise comparisons using the Holm-Šídák test. Error bars indicate SEM. N = 13–16 animals per genotype.

exposed to hypoxia using a published nitrogen displacement approach (*Pocock and Hobert, 2008*; *Park et al., 2012*). As previously described (*Park et al., 2012*), wild-type animals exposed to 0.5% oxygen (hypoxia) localized GLR-1::GFP to elongated structures along dendrites (*Figure 3A,B,E,F*), similar to those observed in *pmk-1* and *sek-1* mutants under normoxic conditions (*Figure 3C,E,F*). These internalized structures can be enhanced in an additive effect when mutations in distinct membrane trafficking pathways are combined in double mutant combinations (*Park et al., 2009*; *Glodowski et al., 2007*; *Shi et al., 2010*). However, we observed no statistically significant additive effect of hypoxia exposure on GLR-1 localization in either *pmk-1* or *sek-1* mutants (*Figure 3D,E,F*). Mutations in *egl-9* under normoxia cause a similar effect on GLR-1 trafficking to that in wild-type animals under hypoxia (*Figure 3E,F*). We therefore examined GLR-1::GFP in double mutants between *egl-9* and either *sek-1* or *pmk-1*. We found no statistically significant difference between the double mutants and either single mutant, under conditions of both normoxia and hypoxia (*Figure 3E,F*). Our results indicate that there is no additive effect of combining mutations that impair p38 MAPK signaling with *egl-9* mutations, hypoxia exposure, or both, suggesting that these factors work together in a single pathway to regulate GLR-1.

EGL-9 and hypoxia regulate multiple physiological processes through their regulation of HIF-1 function (*Shen et al., 2006*; *Chang and Bargmann, 2008*; *Pocock and Hobert, 2008*; *Shao et al., 2010*; *Gort et al., 2008*). However, the regulation of GLR-1 trafficking by EGL-9 and hypoxia does not require HIF-1 (*Park et al., 2012*). Using a *hif-1* molecular null allele (*Jiang et al., 2001*), we tested whether *hif-1* mutations suppressed the accumulation of GLR-1 observed in *sek-1* and *pmk-1* mutants; however, we observed no difference in either *sek-1 hif-1* or *pmk-1 hif-1* double mutants compared to *sek-1* and *pmk-1* single mutants (*Figure 3E,F*). Taken together, our results are consistent with the p38 MAPK components SEK-1 and PMK-1 regulating GLR-1 trafficking through an EGL-9-dependent, HIF-1-independent mechanism.

To examine PMK-1 subcellular localization in these neurons, we generated a transgene containing the *glr-1* promoter driving a full length PMK-1::GFP chimeric protein. We found that PMK-1::GFP was enriched in the nuclei of the command interneurons and distributed in a diffuse fashion throughout the ventral cord dendrites (*Figure 3G*). We also examined animals carrying the same transgene under conditions of hypoxia. In response to hypoxia, we observed a decrease in PMK-1::GFP in both the cell bodies and the dendrites (*Figure 3H*). PMK-1 was less enriched in the nuclei and more diffusely distributed in the cell body cytosol (*Figure 3H,I*). Given that PMK-1 is under the control of the *glr-1* promoter and the *unc-54* 3'UTR sequences in this experiment, and that these regulatory sequences have not shown oxygen-dependent regulation in previous experiments (*Park et al., 2012*; *Ghose et al., 2013*), our results suggest that oxygen elevates PMK-1 levels through a post-transcriptional mechanism.

## CDK-5 acts downstream of p38 MAPK signaling to regulate GLR-1 localization

Under normoxic conditions, EGL-9 promotes GLR-1 recycling by binding to the N-terminus of LIN-10, thereby preventing the kinase CDK-5 from phosphorylating LIN-10 and triggering its diffusion (delocalization) along dendrites (*Park et al., 2012*). If PMK-1 and SEK-1 regulate GLR-1 in the same manner as does EGL-9, then one would expect that (1) a *cdk-5* mutation would suppress the accumulation of GLR-1 observed in *sek-1* and *pmk-1* mutants (similar to how it suppresses accumulation in *egl-9* mutants), and (2) LIN-10 would be diffusely distributed in *sek-1* and *pmk-1* mutants (similar to how LIN-10 is diffusely distributed in *egl-9* mutants) (*Park et al., 2012*). To test the first expectation, we examined GLR-1::GFP in double mutants between *cdk-5* and either *sek-1* or *pmk-1*, and we found that GLR-1 did not accumulate in both double mutants (*Figure 4A–F*). Consistent with the GLR-1 trafficking data, we observed that a *cdk-5* mutation suppressed the spontaneous reversal defects caused by *sek-1* and *pmk-1* mutations (*Figure 4G*). These findings place CDK-5 genetically downstream of SEK-1 and PMK-1.

To test the second prediction, we examined LIN-10 localization in p38 MAPK signaling mutants by introducing a transgene that expresses a functional LIN-10::GFP chimeric protein solely in the GLR-1-expressing command interneurons (*Rongo et al., 1998*). LIN-10::GFP is localized to small puncta along dendrites (*Figure 5A*), and this localization requires EGL-9 and oxygen but is inhibited by CDK-5 (*Park et al., 2012*; *Juo et al., 2007*). We found that LIN-10::GFP was diffusely distributed throughout dendrites (and with few puncta) in *sek-1* and *pmk-1* mutants, similar to its distribution in

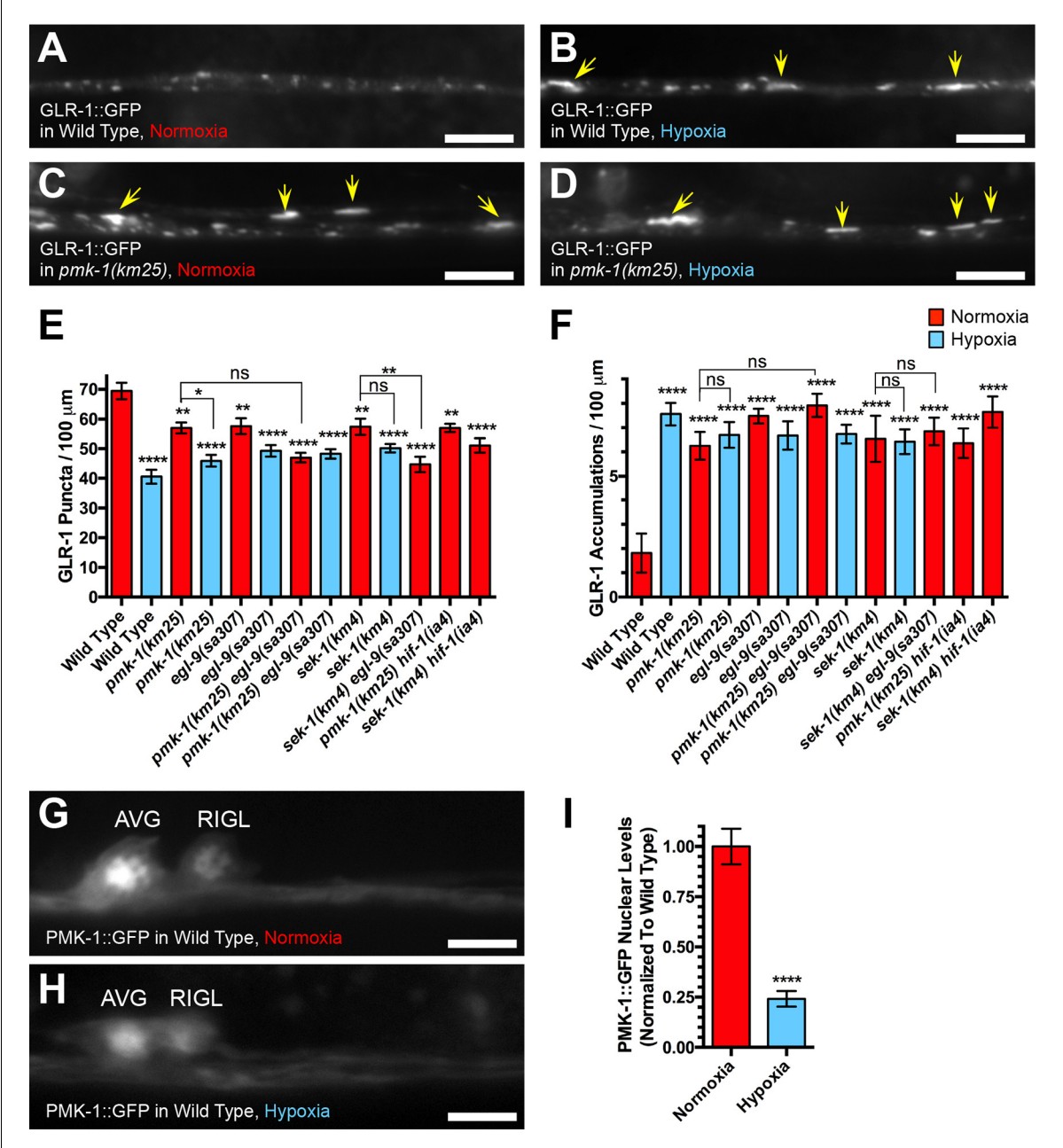

**Figure 3.** Loss of p38 MAPK signaling occludes the effects of hypoxia on GLR-1 AMPAR trafficking. GLR-1::GFP fluorescence in (**A, B**) wild-type animals or (**C, D**) *pmk-1(km25)* mutants under conditions of (**A, C**) normoxia or (**B, D**) hypoxia. Yellow arrows indicate elongated accumulations. Bar: 5 μm. Average GLR-1::GFP number is quantified as (**E**) puncta or (**F**) accumulations per length of ventral cord dendrites. (**G, H**) PMK-1::GFP fluorescence in wild-type animals under (**G**) normoxia or (**H**) hypoxia. Cell bodies for AVG and RIGL are indicated. Average nuclear PMK-1::GFP fluorescence intensity (normalized to the average value in wild type) is quantified in (**I**). Red bar columns indicate animals under normoxia, whereas blue bar columns indicate animals exposed to hypoxia. Graph bar columns labeled with asterisks (****p<0.0001, **p<0.01, *p<0.05) indicate statistical difference by (**E, F**) ANOVA followed by Dunnett's multiple comparison to wild type or Tukey's multiple comparison indicated by the brackets, and (**I**) Student t test. Error bars indicate SEM. N = 11–24 animals per genotype.

*egl-9* mutants and animals undergoing hypoxic stress (***Figure 5B,E***). By contrast, mutations in *cdk-5* result in more LIN-10::GFP puncta (***Figure 5C,E***), and these puncta are larger and brighter, resulting in more total localized LIN-10 along dendrites, which can be quantified as integrated optical density (IOD) along the dendrites (***Figure 5F***). Moreover, the observed delocalization of LIN-10 in *sek-1* and

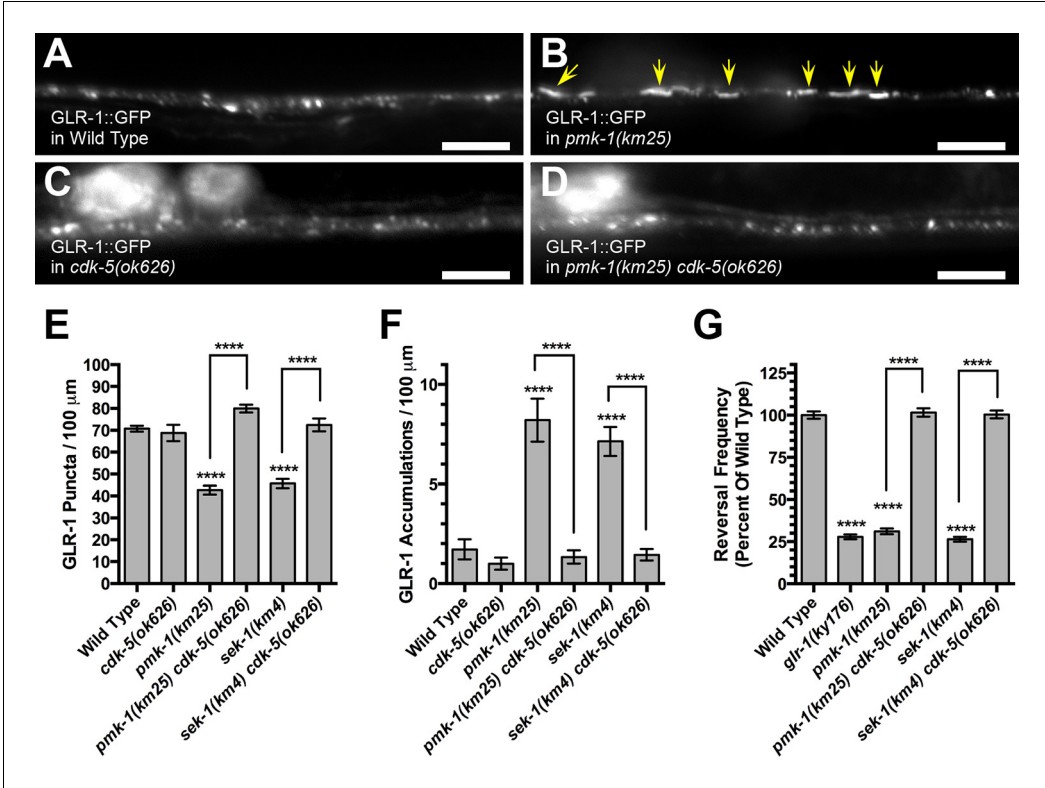

**Figure 4.** CDK-5 is required for p38 MAPK to regulate GLR-1 AMPAR trafficking. GLR-1::GFP fluorescence in (A) wild-type animals, (B) *pmk-1(km25)* mutants, (C) *cdk-5(ok626)* mutants, and (D) *pmk-1(km25) cdk-5(ok626)* double mutants. Yellow arrows indicate elongated accumulations. Bar: 5 μm. Average GLR-1::GFP number is quantified as (E) puncta or (F) accumulations per length of ventral cord dendrites. (G) Spontaneous reversal frequency (number of reversals measured over a 5-min period) represented as a percentage of the mean value for wild type. Graph bar columns labeled with asterisks indicate statistical difference by ANOVA followed by Dunnett's multiple comparison to wild type (****p<0.0001). Lines connecting specific columns indicate pairwise comparisons using the Holm-Šídák test. Error bars indicate SEM. N = 15–28 animals per genotype.

*pmk-1* mutants is completely blocked when a *cdk-5* mutation is introduced into these genetic backgrounds (*Figure 5D,E,F*). This suggests that p38 MAPK signaling promotes LIN-10 localization into puncta by antagonizing the diffuse distribution (delocalization) that would otherwise be promoted by CDK-5. Taken together, our results are consistent with p38 MAPK signaling working together with EGL-9 to promote LIN-10 localization into puncta, and that the underlying mechanism is through the prevention of CDK-5 from opposing LIN-10 localization.

## The p38 MAPK pathway regulates EGL-9 subcellular localization

EGL-9 and oxygen promote LIN-10 subcellular localization, and a specific splice isoform of EGL-9, called EGL-9E, is colocalized with LIN-10 in dendrites (*Park et al., 2012*). As the p38 MAPK pathway could regulate LIN-10 either directly or indirectly by regulating EGL-9, we examined EGL-9E::GFP subcellular localization in p38 MAPK signaling mutants. Whereas EGL-9E::GFP is localized to puncta along the ventral cord dendrites (*Figure 6A,E*), it was diffusely distributed in *sek-1* and *pmk-1* mutants, with few puncta and little total punctate EGL-9E::GFP along dendrites (*Figure 6B,E,F*). One explanation for the impaired localization of EGL-9E in p38 MAPK signaling mutants is if EGL-9E were to depend on LIN-10, the localization of which is also affected in these p38 MAPK mutants. The loss of EGL-9E would thus be a secondary consequence of impaired LIN-10 localization in these mutants. We examined EGL-9E::GFP in *cdk-5* mutants (which have augmented LIN-10 subcellular localization) and double mutants between *cdk-5* and either *sek-1* or *pmk-1*. EGL-9E is localized similar to wild type in *cdk-5* single mutants (*Figure 6C,E,F*), and mutations in *cdk-5* do not suppress the

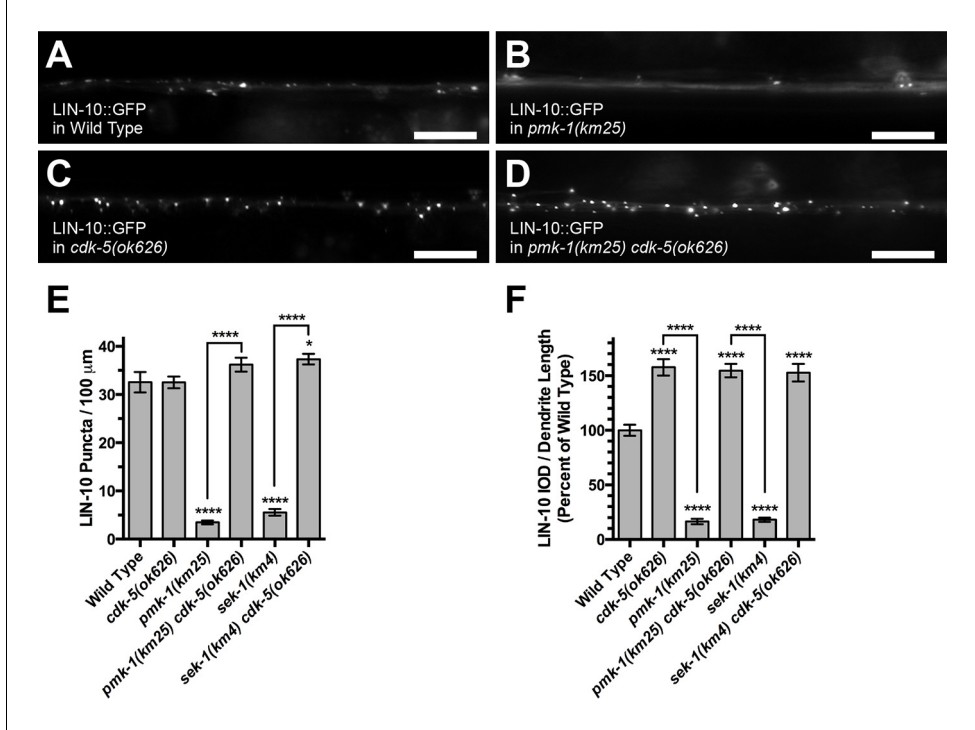

**Figure 5.** The PMK-1 p38 MAPK regulates LIN-10 localization. LIN-10::GFP fluorescence in (**A**) wild-type animals, (**B**) *pmk-1(km25)* mutants, (**C**) *cdk-5(ok626)* mutants, and (**D**) *pmk-1(km25) cdk-5(ok626)* double mutants. Bar: 5 μm. (**E**) Average LIN-10::GFP puncta number is quantified per length of ventral cord dendrites. (**F**) Average integrated optical density (IOD) per puncta per animal as a measurement of total localized LIN-10::GFP. IOD is the sum of the pixel values for each puncta, reflecting both puncta size and fluorescence intensity. Graph bar columns labeled with asterisks indicate statistical difference by ANOVA followed by Dunnett's multiple comparison to wild type (****p<0.0001, *p<0.05). Lines connecting specific columns indicate pairwise comparisons using the Holm-Šídák test. Error bars indicate SEM. N = 15 animals per genotype.

effects on EGL-9E subcellular localization observed in *sek-1* or *pmk-1* mutants (**Figure 6D,E,F**). Similarly, whereas LIN-10 subcellular localization depends on EGL-9E, the subcellular localization of EGL-9E does not require LIN-10 (**Figure 6E,F**). Taken together, our data indicate that p38 MAPK signaling acts genetically upstream of EGL-9, promoting the subcellular localization of the EGL-9E isoform in neurons independent from its effects on LIN-10 or CDK-5.

## The p38 MAPK pathway inhibits the hypoxia response

Given our finding that p38 MAPK signaling promotes EGL-9E subcellular localization and its non-canonical, HIF-1-independent function in the command interneurons, we reasoned that p38 MAPK signaling might also promote canonical EGL-9 function, including its ability to repress HIF-1, throughout the organism. We tested this hypothesis several ways. First, we examined the expression of a transcriptional reporter for HIF-1 using *nIs470*, a transgene containing the *cysl-2* promoter and Venus (**Ma et al., 2012**). Expression from the *cysl-2* reporter is inactive under normoxia but activated by hypoxia (**Figure 7A,B**). We introduced *nIs470* into *pmk-1* mutants and found that *cysl-2* reporter expression under normoxia was elevated in these mutants, similar to the expression observed in wild-type animals under hypoxia (**Figure 7C**). Exposure to hypoxia did not result in increased *cysl-2* reporter expression in *pmk-1* mutants, suggesting that *pmk-1* mutations occlude any additional effects of hypoxia on *cysl-2* transcription (**Figure 7D**).

We also measured the levels of a different HIF-1 target gene, called *nhr-57*, using qRT-PCR to measure endogenous mRNA levels in total nematode lysates (**Shen et al., 2005**). We found that mutations in *sek-1* and *pmk-1* resulted in a seven-fold increase in *nhr-57* mRNA levels relative to a control transcript (actin) (**Figure 7E**), similar to the increase observed in wild-type animals treated

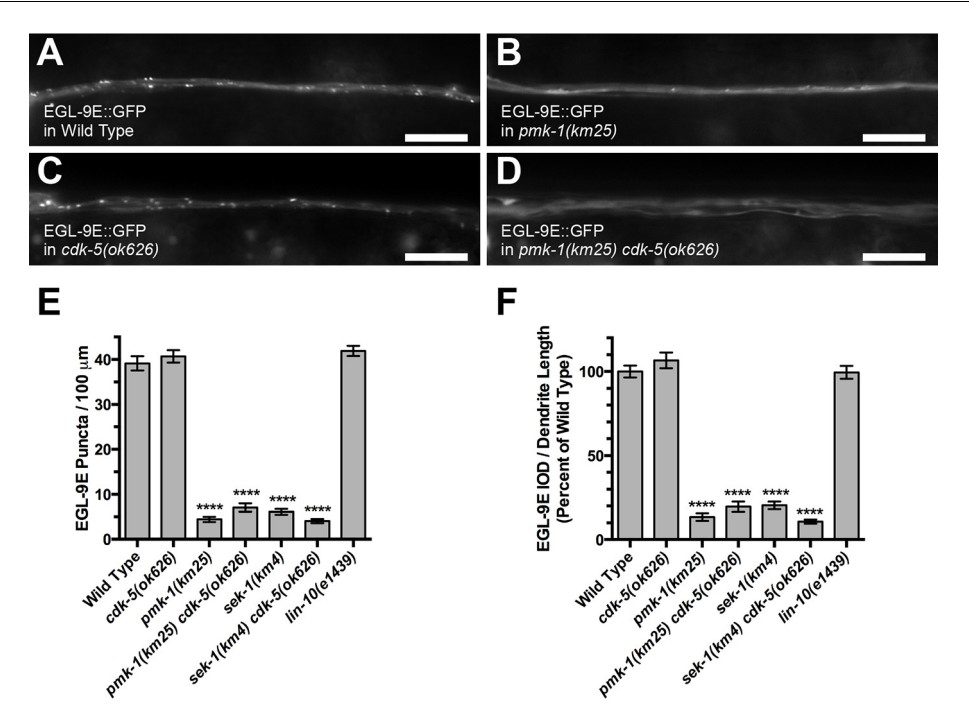

**Figure 6.** The PMK-1 p38 MAPK regulates EGL-9 localization. EGL-9E::GFP fluorescence in (**A**) wild-type animals, (**B**) *pmk-1(km25)* mutants, (**C**) *cdk-5(ok626)* mutants, and (**D**) *pmk-1(km25) cdk-5(ok626)* double mutants. Bar: 5 μm. (**E**) Average EGL-9E::GFP puncta number is quantified per length of ventral cord dendrites. (**F**) Average integrated optical density (IOD) per puncta per animal as a measurement of total localized EGL-9E::GFP. IOD is the sum of the pixel values for each puncta, reflecting both puncta size and fluorescence intensity. Graph bar columns labeled with asterisks indicate statistical difference by ANOVA followed by Dunnett's multiple comparison to wild type (****$p<0.0001$). Error bars indicate SEM. N = 13–20 animals per genotype.

under hypoxia (*Bishop et al., 2004*). HIF-1 is required for this increase, as mutations in *hif-1* blocked the effect of mutations in *sek-1* and *pmk-1* (*Figure 7E*).

As a direct measure of EGL-9 activity, we also visualized its HIF-1 substrate using a transgenic HIF-1::GFP chimeric protein expressed in the command interneurons via the *glr-1* promoter (*Park et al., 2012*). Low levels of HIF-1::GFP are visible in wild-type neurons (*Figure 7F*). By contrast, strong HIF-1::GFP foci are visible in the nuclei of *egl-9* mutants (*Figure 7G*) as well as in *pmk-1* mutants (*Figure 7H*), suggesting that p38 MAPK signaling, like EGL-9, promotes HIF-1 turnover (quantified in *Figure 7I*). Taken together, our results suggest that p38 MAPK signaling modulates the canonical hypoxia response pathway as well as the non-canonical pathway that regulates AMPAR recycling in neurons.

## Age-onset downregulation of GLR-1 AMPARs through p38 MAPK and CDK-5 signaling

The levels of activated PMK-1 decrease as animals age, and this depression in p38 MAPK activity can be detected on Western blots using an anti-phospho-p38 MAPK antibody (*Youngman et al., 2011*). To confirm this change in p38 MAPK signaling, we generated lysates (in three separate biological replicates) from wild-type animals and *pmk-1* mutants either from young larvae (stage L4) or older adults (day 9 post-L4 stage), separated the proteins by SDS-PAGE, and probed them with an anti-phospho-p38 MAPK antibody and an anti-actin antibody as a loading control (*Figure 8A*). We detected a 50% decrease in phospho-PMK-1 levels in older animals relative to young larvae, consistent with a decrease in p38 MAPK signaling during aging (*Figure 8A,B*). We also detected a 30% decrease in *pmk-1* mRNA levels in older animals relative to young larvae (*Figure 8C*). Signaling through PMK-1 and its downstream transcriptional effector ATF-7 promotes the transcription of

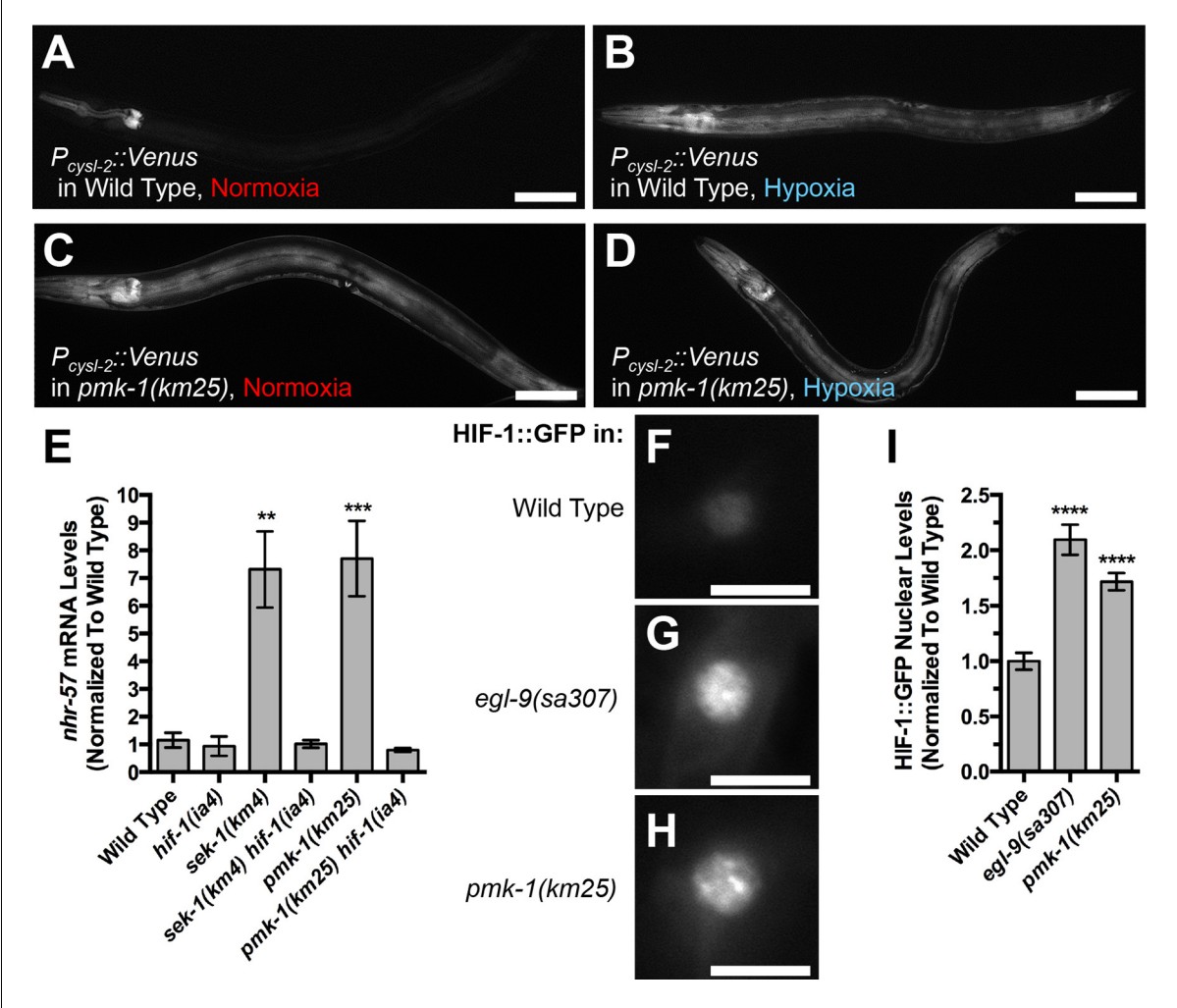

**Figure 7.** The p38 MAPK pathway modulates the hypoxia response pathway. Fluorescence from Venus expressed from the *cysl-2* promoter in animals carrying a $P_{cysl-2}$::*Venus* transgene. Either (**A, B**) wild-type animals or (**C, D**) *pmk-1(km25)* mutants under (**A, C**) normoxia or (**B, D**) hypoxia are shown. Note that pharyngeal fluorescence is detected from the $P_{myo-2}$::mCherry injection marker even under normoxia. Bar: 100 µm. (**E**) Relative *nhr-57* mRNA levels from the indicated genotypes (under normoxia) quantified by qRT-PCR and normalized to the mean value for wild type. (**F, G, H**) Fluorescence from a HIF-1::GFP chimeric protein expressed from the *glr-1* promoter in animals under normoxia and carrying a $P_{glr-1}$::*HIF-1::GFP* transgene. The PVC neuron cell body from (**F**) wild type, (**G**) *egl-9(sa307)* mutants, and (**H**) *pmk-1(km25)* mutants is shown. (**I**) Average relative HIF-1::GFP fluorescence levels (normalized to the mean value for wild type) observed in PVC nuclei under normoxia. Graph bar columns labeled with asterisks indicate statistical difference by ANOVA followed by Dunnett's multiple comparison to wild type (****p<0.0001, ***p<0.001, **p<0.01). Error bars indicate SEM. N = 17–20 animals per genotype.

multiple genes, including that of T24B8.5 (*Shivers et al., 2010*). Thus, a transgene (*agls219*) containing the T24B8.5 promoter driving GFP expression provides an additional means to monitor p38 MAPK signaling via PMK-1 (*Shivers et al., 2009*). We examined GFP expression from the T24B8.5 promoter in L4 stage animals and day 9 adults, finding a baseline GFP expression level in L4 animals (*Figure 8D*) that disappeared in older animals (*Figure 8E*). Expression of GFP from the T24B8.5 promoter was abolished in *pmk-1* mutants at both stages of development (*Figure 8F,G*). Taken together, these findings confirm that PMK-1 p38 MAPK activity wanes in aging animals.

Given our finding that PMK-1 promotes GLR-1 recycling in young adult animals, could an age-dependent decrease in p38 MAPK activity result in GLR-1 internal accumulation in older animals? We tested this possibility by examining GLR-1::GFP in both L4 larvae and in older animals. Whereas L4 larvae resembled young adults, with ventral cord dendrites containing punctate GLR-1::GFP (*Figure 9A,G*), we began to observe GLR-1 in elongated structures along the ventral cord of many

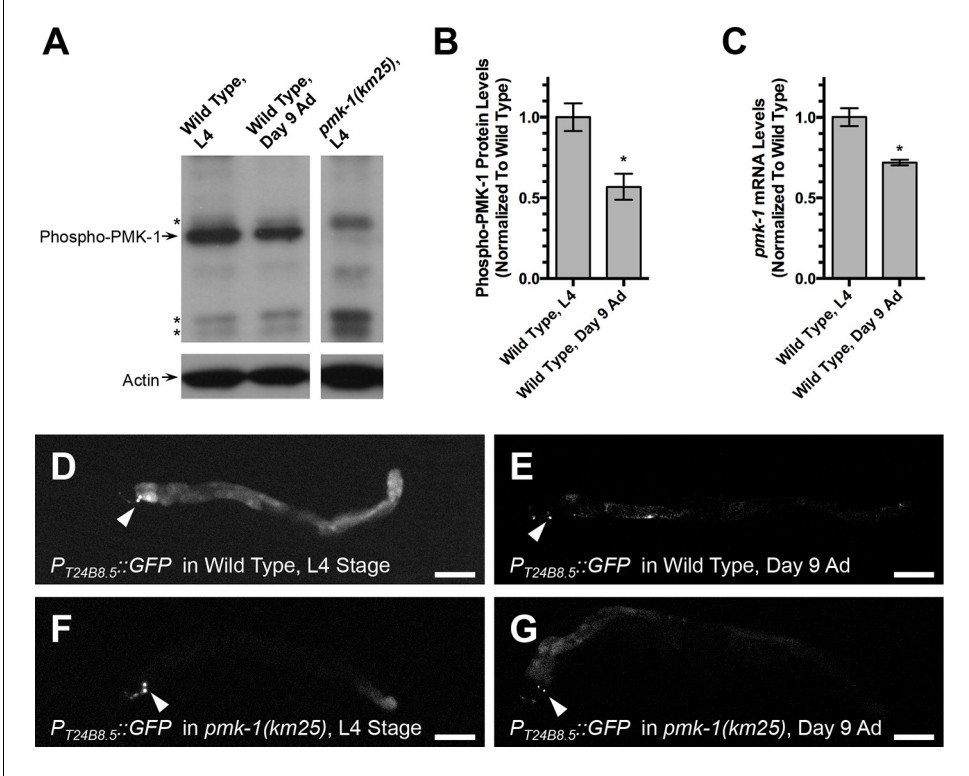

**Figure 8.** PMK-1 p38 MAPK activity declines with aAge. (**A**) Western blot of whole animal lysates from the indicated genotype and developmental stage (either L4 larvae or adults aged 9 days past L4). Top panels show signal from anti-phospho-p38 MAPK antibody, whereas the bottom panels show signal from an anti-actin antibody. Arrows point to bands corresponding to the indicated protein. Asterisks indicate additional bands that cross react with the anti-phospho-p38 MAPK antibody but are not actually PMK-1 (i.e., they are not present in the *pmk-1* molecular null mutant). The panels on the left and the panels on the right are from the same Western blot, but from different regions of the SDS-PAGE gel. (**B**) Quantification of the ratio of anti-phospho-p38 MAPK antibody signal to anti-actin antibody signal, normalized to the value of wild-type L4 animals. Values indicate an average from three independent Western blots. (**C**) Quantification of the ratio of *pmk-1* mRNA to actin mRNA, normalized to the value of wild-type L4 animals. Values indicate an average from three independent qRT-PCR reactions. Graph bar columns labeled with asterisks indicate statistical significance by Student t test (*p<0.05). Error bars indicate SEM. (**D, E, F, G**) Fluorescence from GFP expressed from the *T24B8.5* promoter in animals carrying a $P_{T24B8.5}::GFP$ transgene. Either (**D, E**) wild-type animals or (**F, G**) *pmk-1(km25)* mutants as (**D, F**) L4 stage larvae or (**E, G**) adults 9 days following the L4 stage are shown. Note that GFP expression from AIY (arrowheads) is from a transgenic marker ($P_{ttx-3}::GFP$) incorporated into the array. Bar: 100 μm.

4-day adult (4 days after L4 stage) animals. By day 9 after L4 stage (9-day adults), all older adults had accumulated GLR-1::GFP in elongated accumulations and had fewer GLR-1 puncta, a phenotype similar to that observed in *egl-9, pmk-1*, and *sek-1* mutants (**Figure 9B,G,H**). Consistent with this change in GLR-1 localization, we observed a decrease in spontaneous reversal frequency each day wild-type animals grew older (**Figure 9I**). We also examined 9-day adult *pmk-1* mutants and found that they had similar GLR-1::GFP localization and spontaneous reversal phenotypes to those observed in wild-type adults at all ages (**Figure 9G,H,I**), suggesting that *pmk-1* mutations occlude any additional effect on GLR-1 trafficking and function due to aging. These findings are consistent with depressed p38 MAPK activity causing GLR-1 trafficking defects in older animals.

If older animals accumulate endosomal GLR-1 because of depressed p38 MAPK signaling, then one might expect that (1) overexpressing wild-type PMK-1, or (2) removing CDK-5 activity (the inhibitory target of PMK-1/EGL-9 regulation) might suppress this age-onset phenotype. We tested these possibilities by first generating a transgene that overexpresses a wild-type PMK-1 cDNA via the *glr-1* promoter. Wild-type animals that overexpress PMK-1 from this transgene showed normal punctate

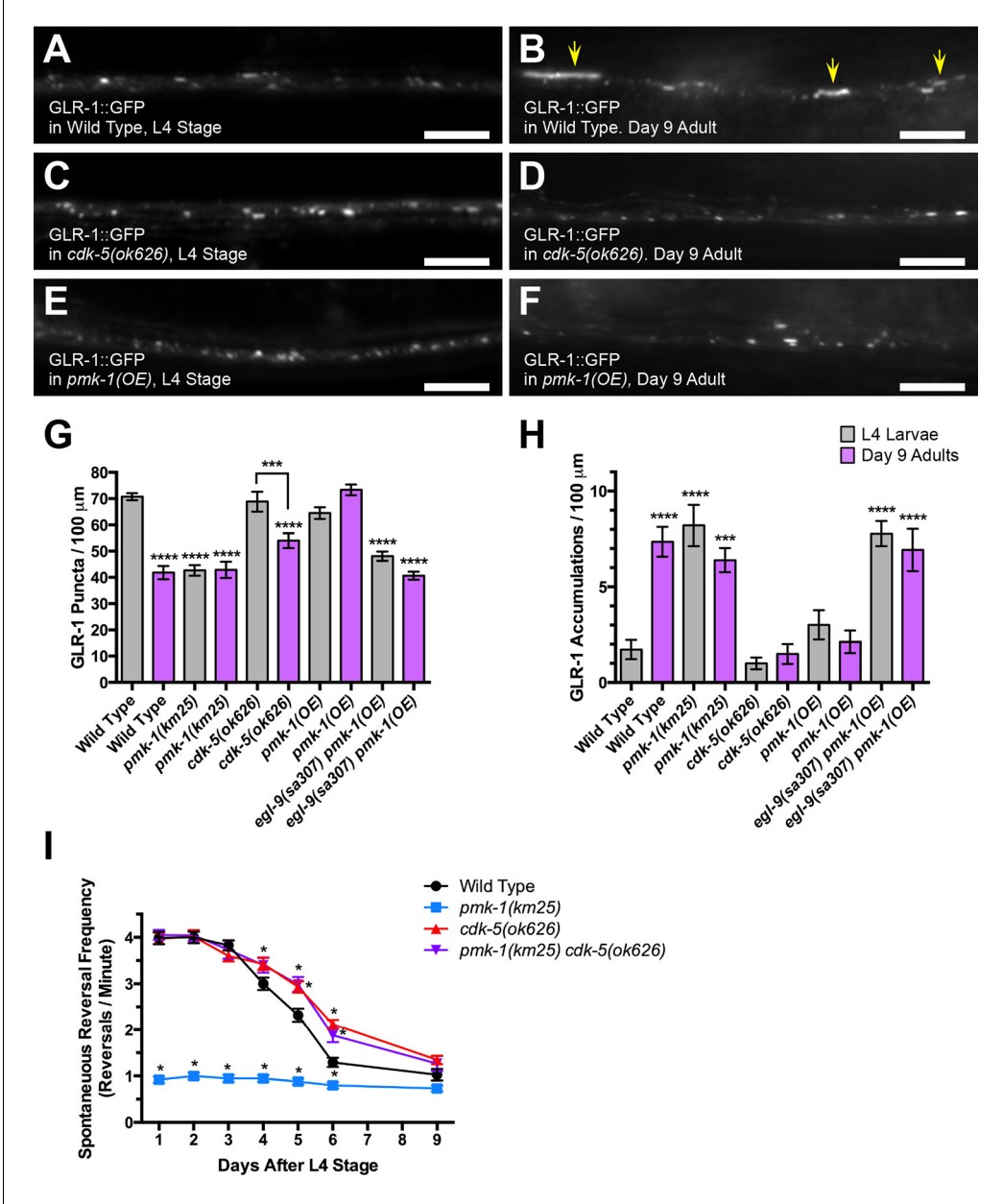

**Figure 9.** Age-onset downregulation of GLR-1 AMPARs through p38 MAPK and CDK-5 signaling. GLR-1::GFP fluorescence in (**A, B**) wild-type animals, (**C, D**) *cdk-5(ok626)* mutants, and (**E, F**) wild-type animals expressing a wild-type *pmk-1* cDNA from the *glr-1* promoter (from a P*glr-1*::PMK-1(+) transgene, labeled as *pmk-1(OE)* to indicate PMK-1 overexpression). Animals are either (**A, C, E**) L4 stage larvae or (**B, D, F**) adults aged 9 days past the L4 stage. Yellow arrows indicate elongated accumulations. Bar: 5 μm. Average GLR-1::GFP number is quantified as (**G**) puncta or (**H**) accumulations per length of ventral cord dendrites. Gray bar columns indicate L4 stage animals, whereas purple bar columns indicate older animals that are 9 days past the L4 stage. Graph bar columns labeled with asterisks indicate statistical difference by ANOVA followed by Dunnett's multiple comparison to wild type (****$p < 0.0001$, ***$p < 0.001$). Lines connecting specific columns indicate pairwise comparisons using the Holm-Šídák test. Error bars indicate SEM. N = 13–18 animals per genotype. (**I**) Spontaneous reversal frequency (number of reversals measured over a 5-min period and normalized per minute) as measured at different days after the L4 stage in aging animals of the indicated genotype. Asterisks indicate statistical difference by ANOVA followed by Dunnett's multiple comparison to wild type (*$p < 0.05$) at the indicated time point.

GLR-1::GFP even in 9-day adults (*Figure 9E–H*), indicating that simply elevating PMK-1 is sufficient to suppress the defects observed during aging (when the levels of activated PMK-1 are observed to drop). Next, we examined older *cdk-5* mutants and found that they also failed to accumulate endosomal GLR-1 (*Figure 9C,D,G,H*), indicating that CDK-5 is required for the defect in GLR-1 recycling in older animals. Mutations in *cdk-5* also partially restored GLR-1 function in aging animals, as *cdk-5* adults as late as 6 days post L4 have elevated rates of spontaneous reversals relative to wild-type animals of the same age (*Figure 9I*). Moreover, mutations in *cdk-5* restore reversals to *pmk-1* mutants in young and old animals to levels that were higher than those for wild-type animals (*Figure 9I*). Whereas *cdk-5* mutants showed more robust reversal behavior during aging relative to wild type, they nevertheless still showed a decline in reversals and GLR-1 puncta number over time, suggesting that additional factors contribute to age-associated decline in this behavioral modality. For example, muscle decline dramatically impairs locomotion in 9-day adults, making interpretation of more subtle behavioral phenotypes like reversal frequency challenging (*Herndon et al., 2002*). Taken together, our results indicate that AMPAR GLR-1 recycling and function decline with age, and that this decline is due, at least in part, to depressed p38 MAPK signaling, subsequent activation of the HIF-1-independent hypoxia response pathway, and CDK-5 activity.

## Discussion

Here we have shown that p38 MAPK signaling modulates both the canonical hypoxia response pathway and a non-canonical hypoxia response pathway that regulates GLR-1 AMPAR trafficking and GLR-1-mediated behavior. The canonical hypoxia pathway senses oxygen via the prolyl hydroxylase EGL-9, which uses dioxygen to hydroxylate a proline residue on the transcription factor HIF-1, resulting in the ubiquitin-mediated degradation of HIF-1 (*Epstein et al., 2001*). The non-canonical pathway senses oxygen via a specific isoform of EGL-9, called EGL-9E, which, when activated by oxygen, binds to the scaffolding molecule LIN-10, recruiting it to endosomes where it promotes the recycling of GLR-1 AMPARs to the synapse (*Park et al., 2012*). We find that if animals are under normal oxygen conditions and are young, then the p38 MAPKK SEK-1 and the p38 MAPK PMK-1 promote EGL-9 activity (*Figure 10A–D*). Active EGL-9 in turn triggers HIF-1 turnover, thereby preventing a HIF-1 transcriptional response. In addition, p38 MAPK signaling also promotes the association of EGL-9E with LIN-10, in turn mediating steady GLR-1 recycling. By contrast, if animals are under conditions of hypoxia, then EGL-9 activity is depressed, resulting in HIF-1 stabilization and the activation of the HIF-1 transcriptional response. In addition, depressed EGL-9 activity exposes LIN-10 to phosphorylation by the CDK-5 kinase, resulting in LIN-10 delocalization in neurons and depressed GLR-1 recycling (*Figure 10E–H*). This regulation results in long term changes in synaptic efficacy in the command circuit, which switches locomotion behavior from local foraging to long-distance roaming. This behavioral switch in locomotion allows the animal to escape the hypoxic environment. As animals grow older, p38 MAPK activity decreases despite the presence of ample oxygen, resulting in impaired GLR-1 recycling through the action of CDK-5. Our findings suggest that p38 MAPK signaling is an important part of the hypoxia response pathway.

*C.elegans* encounters hypoxic and anoxic environments in the soil, which contains bacteria and rotting material (*Anderson and Dusenbery, 1977*; *Van Voorhies and Ward, 2000*). Gas exchange occurs through the cuticle, and oxygen is sensed in the fluid of the pseudocoelomic body cavity by a set of sensory neurons expressing soluble guanylate cyclases; these neurons mediate a rapid aerotaxis response to changes in oxygen levels (*Cheung et al., 2005*; *Gray et al., 2004*). By contrast, the PMK-1/EGL-9/HIF-1 pathway is broadly expressed, activates a slower response to hypoxia that is tailored for cellular stress, modulates the aerotaxis circuit describe above, and modulates the locomotory reversal circuit that nematodes use to escape hypoxic environments (*Chang and Bargmann, 2008*; *Pocock and Hobert, 2008*; *Branicky and Schafer, 2008*; *Park et al., 2012*). The two pathways are integrating environmental information on oxygen availability over different time scales: the former over a period of seconds to minutes whereas the latter is over a period of minutes to hours.

There are precedents for gas messenger-dependent oxygen sensing mechanisms that act through protein kinases. In the mammalian carotid body, oxygen stimulates heme oxygenase-2 to generate carbon monoxide (CO), which activates soluble guanylate kinase and protein kinase G (PKG) (*Yuan et al., 2015*; *Peng et al., 2014*; *Maines, 2004*). PKG inactivates cystathionine-γ-lyase (CSE) through direct phosphorylation, resulting in reduced hydrogen sulfide levels, reduced carotid body

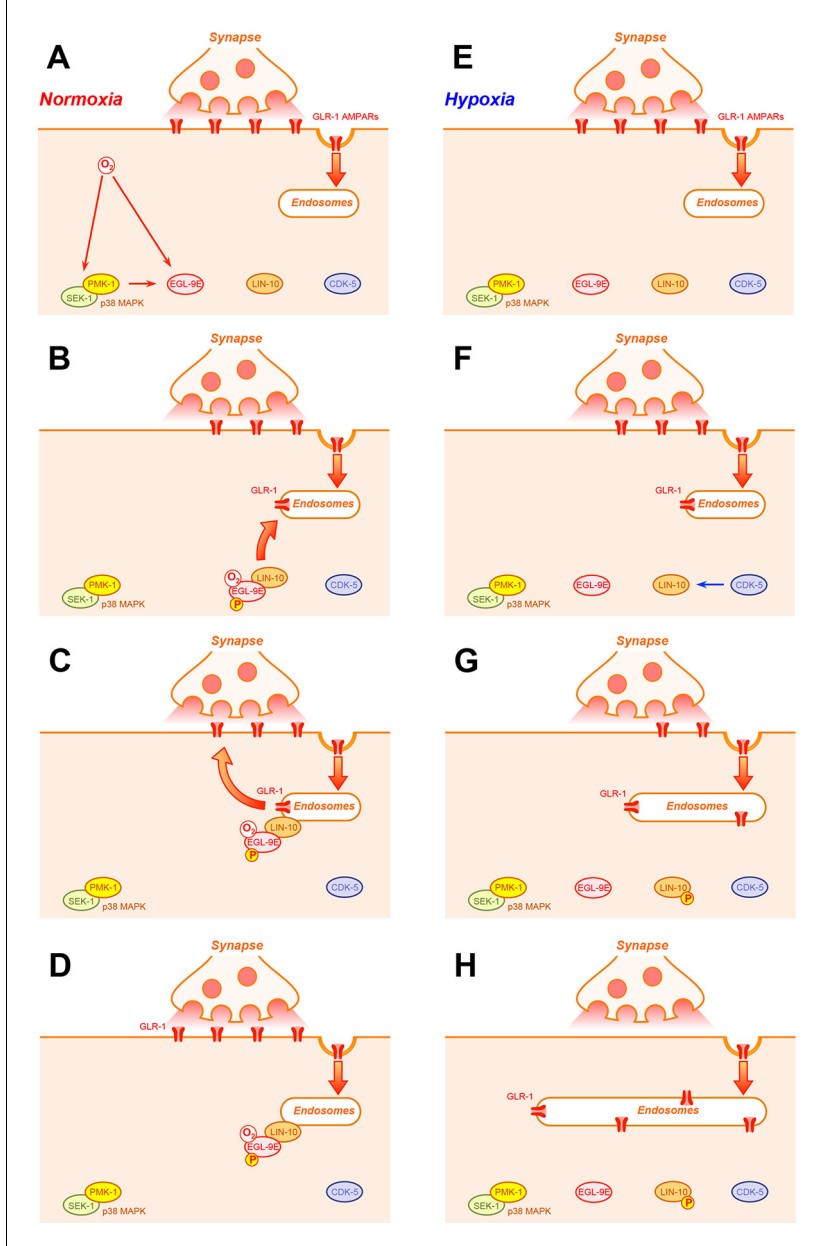

**Figure 10.** Hypothetical model for p38 MAPK regulation of the hypoxia response pathway. A hypothetical, step-by-step model of hypoxia response pathway interactions in *C. elegans* neurons is shown for conditions of either (**A-D**) normoxia or (**E-H**) hypoxia. (**A**) Under normoxia, oxygen binds to and activates EGL-9 (pink ovals). Oxygen also activates SEK-1 and PMK-1 (p38 MAPK, green and yellow ovals, respectively) through a mechanism that remains unknown. Activated p38 MAPK in turn phosphorylates one or more proteins (possible EGL-9 itself, as speculated in this cartoon with a 'P' in a yellow circle) that activate EGL-9 and trigger its recruitment to endosomes. Meanwhile, GLR-1 receptors (red channels) undergo continual endocytosis at the synapse. (**B**) Isoform EGL-9E, now bound to oxygen and possibly phosphorylated by p38 MAPK, becomes localized to endosomes, where it binds the PDZ-PTB domain protein LIN-10 (orange oval) and recruits it to endosomes by preventing its phosphorylation by the CDK-5 kinase (purple oval). (**C**) Once at endosomes, LIN-10 promotes the recycling of endocytosed GLR-1 AMPARs (red channels in the endosome) back to the synapse. (**D**) The final outcome is that GLR-1 synaptic levels are maintained. (**E**) Under hypoxia, lack of oxygen results in lower SEK-1/PMK-1 p38 MAPK activity and inactive EGL-9. (**F**) In the absence of oxygen, EGL-9E does not bind to LIN-10. This exposes the LIN-10 N-terminus (the localization domain of LIN-10) to CDK-5, which phosphorylates it, thereby inhibiting LIN-10 recruitment to endosomes. (**G**) Without endosomal LIN-10, GLR-1 AMPARs continue undergoing endocytosis from

*Figure 10 continued on next page*

*Figure 10 continued*

synapses but are not recycled, resulting in their accumulation in elongated endosomal compartments. (H) The final outcome is that GLR-1 synaptic levels become depleted.

neural activity, and regular breathing (*Yuan et al., 2015*). Under hypoxia, a drop in CO levels results in inactive PKG, CSE activation, increased hydrogen sulfide ($H_2S$) levels, increased carotid body neural activity, and accelerated breathing.

Changes in oxygen levels are known to regulate p38 MAPK itself. In mammals, hypoxic damage (e.g., triggered by ischemia during myocardial infarction or stroke) can induce the activation of p38 MAPK, which is activated by stress and inflammation (*Cook et al., 1999*; *Kawasaki et al., 1997*). The activation of p38 MAPK contributes to ischemic injury, necrosis, and apoptosis, resulting in heart failure in the case of myocardial infarction and neurodegeneration in the case of ischemic stroke (*Marber et al., 2011*; *Kumphune et al., 2012*; *Barone et al., 2001*; *Lai et al., 2014*). The specific mechanism by which activated p38 MAPK contributes to ischemia-induced damage is unclear, but is thought to stem in part from its regulation of growth factor and apoptosis signal transduction pathways. Several groups have observed changes in HIF activity triggered by ERK and p38 MAPK (possibly activated by ROS generated from mitochondria under hypoxia), yet it remains controversial whether the mechanism is direct phosphorylation of HIF-1 by p38 MAPK (*Conrad et al., 1999*; *Bardos and Ashcroft, 2004*; *Minet et al., 2001*). Our findings here would suggest a novel mechanism by which p38 MAPK signaling regulates HIF-1: the regulation of the PHD enzymes that act upstream of HIF-1. This mechanism could provide an important link between growth factor signaling and the hypoxia response pathway in maintaining stem cell populations and promoting tumor growth and metastasis (*Wang et al., 2013*).

If p38 MAPK regulates HIF-1 activity, then what is the specific physiological role of this regulation? To what is p38 MAPK responding? While PHD proteins are well-established oxygen sensors in the hypoxia response pathway, there is also a likely role for the mitochondrial electron transport chain (ETC) in sensing oxygen during hypoxia. Even under conditions of normal oxygen, the ETC produces low levels of ROS (*Turrens, 2003*). Perhaps ROS activates a baseline level of p38 MAPK under normoxic conditions, acting as part of an additional oxygen sensing mechanism (*Figure 10A*). Indeed, we did observe that normal oxygen levels promoted PMK-1 nuclear localization in the command interneurons, and hypoxia resulted in PMK-1 depletion from the nucleus, suggesting that oxygen (or its byproducts) could be activating PMK-1 (*Figure 3G,H,I*).

An alternative explanation for why we observe activation of the hypoxia response pathway in *pmk-1* mutants is that the activation is due to an indirect effect of losing the baseline expression of the oxidative stress response in these mutants. In this scenario, *pmk-1* mutants accumulate products of oxidative stress (e.g., ROS and oxidized macromolecules), which would mimic hypoxia, perhaps through the actions of ROS directly inactivating EGL-9. We feel that this scenario is unlikely because *skn-1* mutants, which are arguably more impaired for the oxidative stress response than are *pmk-1* mutants, do not show induction of hypoxia response target genes like *nhr-57* (*Oliveira et al., 2009*). Nor did *skn-1* mutants show the same GLR-1 localization defects that we observed in *pmk-1* and *egl-9* mutants. Instead, we favor a model in which PMK-1 directly silences the hypoxia response during normoxia, and that it does so independent of its role in the oxidative stress response. It remains a possibility that the additional ROS that is generated during long-term hypoxia and/or reoxygenation might further activate PMK-1, providing a negative feedback that restores the hypoxia response back to a normoxia baseline, thereby minimizing the dangers of ROS production that occur during extended hypoxia and subsequent reoxygenation. Too much PMK-1 activation (e.g., during extreme anoxia) might contribute to toxicity; indeed, mutations in *pmk-1* increase the survival of animals exposed to long-term anoxia (*Hayakawa et al., 2011*)

Consistent with our model, *C. elegans* PMK-1 is activated by oxidative stress in addition to being activated by pathogenic infection (*Berman et al., 2001*; *Kim et al., 2002*; *Inoue et al., 2005*). Bacterial pathogens and anoxia exposure can both activate PMK-1 through the Toll/IL-1 resistance (TIR) domain protein TIR-1, the ASK1 ortholog MAPKKK NSY-1, and the MKK3 MAPKK SEK-1 (*Liberati et al., 2004*; *Papp et al., 2012*; *Kim et al., 2002*; *Hayakawa et al., 2011*). Whereas SEK-1 and PMK-1 are activated by oxidative stress, NSY-1 and TIR-1 do not appear to be critical

components through which *C. elegans* respond to oxidative stress and presumably ROS (*Inoue et al., 2005*). Our results clearly show that NSY-1 is not required for oxygen levels to modulate the hypoxia response, and given that NSY-1 is the sole *C. elegans* ASK1 ortholog, it seems likely that SEK-1 and PMK-1 are activated by a different MAPKKK under these conditions. Identifying the specific MAPKKK will be an important next step in determining how the p38 MAPK PMK-1 pathway senses hypoxia (perhaps through an alternative ROS sensor) and modulates the hypoxia response.

In addition to its acute role in promoting survival during oxygen deprivation stress, the hypoxia response pathway also has a complex role in regulating aging and lifespan beyond simply maintaining stem cell populations (*Katschinski, 2006*). In *C. elegans*, hypoxia and limited stabilization of HIF-1 promote longevity (*Leiser and Kaeberlein, 2010*). However, loss of EGL-9, which results in extreme HIF-1 stabilization, does not promote longevity and can be actually detrimental to lifespan (*Chen et al., 2009*; *Bellier et al., 2009*). Moreover, loss of HIF-1 can also promote longevity under conditions of elevated temperature via a separate mechanism (*Leiser et al., 2011*). As nematodes grow older, the levels of active PMK-1 decrease (*Youngman et al., 2011*), which might result in elevated HIF-1 activity. Whereas mutants for *pmk-1* have a similar lifespan to that of wild type (*Troemel et al., 2006*; *Alper et al., 2010*), it is worth noting that the observed decrease in PMK-1 levels over time could impair nervous system function in a manner that would be missed by simple life span analysis. We find that GLR-1 accumulates in endosomes as nematodes grow older, similar to what occurs in young *pmk-1* mutants, and that either the simple overexpression of PMK-1 or the removal of CDK-5, an inhibitor of its downstream target LIN-10, restores both GLR-1 localization and function to levels observed in younger animals. Our findings highlight the idea that changes in kinase signaling could explain aspects of age-associated physiological decline.

## Materials and methods

### Transgenes and Germline Transformation

Transgenes generated in this study include (1) a wild-type *pmk-1* cDNA fused to the *glr-1* promoter, (2) a wild-type *sek-1* cDNA fused to the *glr-1* promoter, and (3) a wild-type *pmk-1* cDNA fused in frame to GFP and placed behind the *glr-1* promoter. Transgenic plasmids were generated using standard techniques. Transgenic strains were isolated after microinjecting plasmids (10 ng/µl) with the transgenic marker *ttx-3::rfp* (a gift from O. Hobert, Columbia Univ.) into the germline to form extrachromosomal arrays. All other transgenes used in this study were as described in the publications cited in the text.

### Hypoxic exposure

Animals were grown at 20°C on standard NGM plates seeded with OP50 *E. coli*. For hypoxia, animals were incubated in a hypoxia chamber (C-174 chamber, Biospherix) for 24 hr at 20°C and recovered in ambient oxygen for 12 hr at 20°C. The oxygen level was automatically maintained with an oxygen controller (ProOx P110, Biospherix) supplied with compressed nitrogen gas.

### Fluorescence microscopy

GFP- and RFP-tagged fluorescent proteins were visualized in nematodes by mounting larvae on 2% agarose pads with levamisole. Fluorescent images were observed using a Zeiss Axioplan II. A 100X (N.A. = 1.4) PlanApo objective was used to detect GFP and RFP signals. Imaging was done with an ORCA charge-coupled device (CCD) camera (Hamamatsu, Bridgewater, NJ) using IPLab software (Scanalytics, Inc, Fairfax, VA) or iVision v4.0.11 (Biovision Technologies, Exton, PA) software. Exposure times were chosen to fill the 12-bit dynamic range without saturation. Maximum intensity projections of z-series stacks were obtained and out-of-focus light was removed with a constrained iterative deconvolution algorithm (iVision). For images, we captured the anterior ventral cord dendrites in the anterior region containing the RIG and AVG cell bodies.

The quantification of ventral nerve cord fluorescent objects (i.e., puncta and elongated compartments) was done using ImageJ (*Collins, 2007*) to automatically threshold the images and then determine the outlines of fluorescent objects in ventral cord dendrites. ImageJ was used to quantify both the shape and the size of all individual fluorescent objects along the ventral cord. This allowed us to distinguish between the small GLR-1::GFP puncta in wild-type animals and the large, aberrant

compartments (which have an elongated shape rarely observed in wild type) in hypoxic animals, as well as in thevariousindicated mutants. Object size was measured as the maximum diameter for each outlined puncta. Object number was calculated by counting the average number of puncta per 100 microns of dendrite length. The amount of a given fluorescent protein per puncta was calculated by summing all of the pixel values contained within each individual punctum to yield an integrated optical density (IOD) score for each punctum.

Colocalization between GLR-1::GFP and mRFP::SYX-7 was performed as previously described (*Park et al., 2009*). Single optical images for neuronal cell bodies expressing both reporters were collected using a confocal microscope equipped with the BD CARV II Confocal Imager and a Carl Zeiss 100× Plan-Apochroma objective (N.A. = 1.4). For quantitative colocalization analysis, all image manipulations were performed with iVision v4.0.11 (Biovision Technologies, Exton, PA) software using the FCV colocalization function. We applied an empirically derived threshold to all images for both the GLR-1::GFP channel and the mRFP::SYX-7 channel to eliminate background coverslip fluorescence and other noise (typically 5% of pixels for each channel). The fluorescent intensity values for both the GLR-1::GFP and mRFP::SYX-7 channels were then scatter plotted for each pixel. Pixels with similar intensity values for both channels (within an empirically-established tolerance factor) were counted as colocalized. To acquire the fraction of GLR-1::GFP colocalized with mRFP::SYX-7, the number of colocalized pixels was normalized to the number of GLR-1::GFP pixels under threshold. To maximize our resolving power while observing the relatively small *C. elegans* neuron cell bodies, we restricted our analysis to a single confocal optical section taken through the middle of each cell body.

## Behavioral assays

The reversal frequency of individual animals was assayed as previously described, but with some modifications (*Zheng et al., 1999*). Single young adult hermaphrodites were placed on NGM plates in the absence of food. The animals were allowed to adjust to the plates for 5 min, and the number of spontaneous reversals for each animal was counted over a 5-min period. Twenty or more animals were tested for each genotype, and the reported scores reflect the mean number of reversals per minute, normalized as a percentage of the value of wild-type controls.

## Immunodetection of phospho-PMK-1

To measure phospho-PMK-1 protein levels, 35 young adults of each genotype were dissolved in 1× Laemmli buffer by flash freezing and boiling for 10 min. Lysates were analyzed on 10% SDS-polyacrylamide gels. Western blotting was performed using rabbit anti-phospho-p38 MAPK (1:2000, Promega) and mouse anti-Actin (1:2000, MP biomedicals), with detection through chemiluminescence.

## Real-Time PCR measurements of mRNA levels

Total RNAs were extracted with Trizol (Invitrogen Co., Carlsbad, CA). Young adult or L4 stage larvae (10–15 animals each) were resuspended in 250 µl of Trizol and lysed by one round of freezing (by liquid nitrogen) and thawing (60°C) with subsequent vigorous vortexing in 4°C for 30 min. PCR was performed in an Eco real-time qPCR system (Illumina, San Diego,CA) using iScriptTM One-Step RT-PCR Kit With SYBR Green (Bio-Rad Laboratories Inc., Hercules, CA) in 20 µL reactions with 20 ng of RNA template. For *glr-1*, we used as forward (5′-TGATACAATGAAAGTTGGAGCAAATC-3′) and reverse (5′-CATCGCATTGTCCTCTATCATACCAC-3′) primers. For *pmk-1*, we used as forward (5′- CTGATG-AGCCAATTGCAGAAG-3′) and reverse (5′-TTTTCTCCTCATCTTCCTCTTCG-3′) primers. For *nhr-57*, we used as forward (5′-CGTGATTGCAGACTTGAAAGC-3′) and reverse (5′-GCGTTTGACTTCCATC-GTTTG-3′) primers. For *act-1*, we used as forward (5′-ACCATGTACCCAGGAATTGC-3′) and reverse (5′-TGGAAGGTGGAGAGGGAAG-3′) primers. Samples were measured two to three times and average values were used for the calculation of relative fold changes. The relative levels of *glr-1, pmk-1, and nhr-57* mRNA were normalized to the levels of *act-1* mRNA in each preparation. For each experiment, the value for wild type was set to 1 and other values were normalized accordingly.

## Acknowledgements

We thank the *C. elegans* Genetics Center, Villu Maricq, Joshua Kaplan, Peter Juo, Shohei Mitani, Dennis Kim, Oliver Hobert, Bob Horvitz, and Jo Anne Powell-Coffman for reagents and strains. We thank Piya Ghose for technical assistance with *sek-1* and *pmk-1*. We thank members of the Rongo lab for comments on the manuscript. This work was supported by National Institutes of Health (NIH) R01 NS42023 and R01 GM101972.

# Additional information

### Funding

| Funder | Grant reference number | Author |
| --- | --- | --- |
| National Institutes of Health | R01 NS42023 | Christopher Rongo |
| National Institutes of Health | R01 GM101972 | Christopher Rongo |

The funders had no role in study design, data collection and interpretation, or the decision to submit the work for publication.

### Author contributions

ECP, Designed and performed the genetic, molecular, behavioral, and cell biological experiments, Conception and design, Acquisition of data, Analysis and interpretation of data, Drafting or revising the article; CR, Supervised genetic, molecular, behavioral, and cell biological experiments, Conception and design, Analysis and interpretation of data, Drafting or revising the article

### Author ORCIDs

Eun Chan Park, http://orcid.org/0000-0002-6147-4088
Christopher Rongo, http://orcid.org/0000-0002-1361-5288

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
