## [Decision Letter]

Thank you for submitting your work entitled "The p38 map kinase pathway modulates the hypoxia response and glutamate receptor trafficking in ageing neurons" for consideration by *eLife*. Your article has been evaluated by Eve Marder (Senior Editor) and three reviewers, one of whom, Jan-Marino Ramirez, is a member of our Board of Reviewing Editors, and another is Ganesh Kumar.

The reviewers have discussed the reviews with one another and the Reviewing editor has drafted this decision to help you prepare a revised submission.

Summary:

Combining behavioral and various genetic (knock-ins and single/double knockouts) approaches this study implicates the p38 Map kinase pathway in the hypoxic response of young and old *C. elegans*. The results are timely and of general interest, because increasing evidence from various animal models suggests that the MAPK/P38 kinase pathway is associated with the hypoxic response, oxidative stress, inflammation, and aging, but the underlying mechanisms are often not fully understood. Here, the authors delineate a novel pathway involving components of p38 MAP kinase signaling that participates in the control of glutamate receptor (GLR1) trafficking and GLR-1 mediated reversal behavior. Interestingly, the identified pathway is mediated via an HIF-independent mechanism involving EGL-9, a prolyl hydroxylase isoform. A strength of the study is that authors use multiple assays to measure the interactions between p38 MAPK and hypoxia-sensing including measurements of glutamate receptor distribution within axons, surface-accessibility of glutamate receptors and a behavioral assay that reflects the function of glutamatergic synapses in motor circuits.

Essential revisions:

1) To highlight and contrast this pathway from other published views on oxygen sensing, please consider the following studies. There is some evidence for a role of guanylate cyclase homologue in oxygen sensation in *C. elegans* (Gray et al., Nature 430: 317-322, 2004). Also, there is a recent paper in Science Signaling elaborates a gas messenger-dependent acute oxygen sensing involving carbon monoxide-PKG-mediated phosphorylation of target enzyme as a critical step (Yuan et al., Sci. Signal 8(373): ra37, 2015). A pertinent discussion on these findings in comparison to the current work will further highlight the prevalence of context specific variations in oxygen sensing mechanisms.

2) The Discussion section needs some rearrangement. The importance of the identified O2-ROS-p38 MAP kinase phosphorylation (as shown in Figure 10) of the LIN-1/EGL-9 is lost amongst lengthy discussion on studies showing increased ROS formation under hypoxia by other investigators. Putative role of ROS under hypoxia in the current model of the study is only speculative and therefore can be shortened considerably.

3) Two key questions are posed in Figure 3. First, do PMK-1 and EGL-9 act in a common pathway? Second, is PMK-1 required for the effects of hypoxia on receptor localization? All the data needed to ask these questions are in panels E and F but the comparisons are not. The authors should indicate whether the difference between *pmk-1* mutants, *egl-9* mutants and *pmk-1; egl-9* double mutants can be attributed to chance or not. Ditto for the differences between *pmk-1* mutants raised in normoxic and hypoxic. To put it more simply: the asterisks in panels E and F don't help to make the comparisons.

4) Please add some clarifications to the experiments that measure glutamate receptors in endosomes using co-localization with SYX-7 (Figure 2). First, these experiments determine the distribution of glutamate receptors in compartments in the cell body unlike every other experiment in which receptor distribution is determined in neurites. Is SYX-7 marking endosomes in neurites? If not, then the SYX-7-positive endosomes in the soma differ from the endosomes in neurites and what happens in these endosomes doesn't necessarily inform us about what happens in neurites.

Also, the Methods should make clear how this colocalization was measured. It seems that the data are single optical sections through cell bodies neurons expressing both markers, but this isn't clear in the manuscript. Because the distribution of fluorescence within the cell is complicated and the colocalization between signals is not obvious it is critical that the reader understand how the data are analyzed to reach the conclusion that PMK-1 affects localization to endosomes.

5) In your model (Figure 10), please indicate the phosphorylation state of LIN-1 or EGL-9 under normoxia and hypoxia.

6) One of the reviewers had comments that suggested additional experiments, "low-hanging fruits" that would add considerably to the paper. We are not requiring these for acceptance, but ask that you consider whether you can do some or all of these rapidly, and if so, we would welcome their addition. But, in any case please address these comments.

a) An interesting question is where PMK-1 is localized in these neurons. Is there a pool of p38 MAPK in the neurite that acts locally? The presented model suggests that this is what the authors are thinking, and this idea is readily testable.

b) The authors propose that the p38 MAPK PMK-1 acts in a novel pathway to regulate glutamate receptor localization. This claim is based on the observation that the known upstream MAPKKK for PMK-1, NSY-1, and a known target of PMK-1 signaling, SKN-1, are not required for receptor localization. This is an interesting point and would benefit from additional data. The authors could test other genes linked to PMK-1 signaling, such as TIR-1 and ATF-7. Also, there are a small number of MAPKKKs encoded by the *C. elegans* genome. If the canonical MAPKKK is not regulating PMK-1 in this context perhaps one of the others is. There are a small number of predicted MAPKKKs in *C. elegans*. Do any MAPKKK mutants have defects in glutamate receptor localization?

[Editors' note: further revisions were requested prior to acceptance, as described below.]

Thank you for resubmitting your work entitled "The p38 map kinase pathway modulates the hypoxia response and glutamate receptor trafficking in aging neurons" for further consideration at *eLife*. Your revised article has been favorably evaluated by Eve Marder (Senior editor) and three reviewers, one of whom is a member of our Board of Reviewing Editors. The manuscript has been improved but there are some remaining issues that need to be addressed before acceptance, as outlined below:

“… We generated a transgene containing the glr-1 promoter driving a full length PMK-1::GFP chimeric protein”. *'glr-1'* needs italics.

Throughout the manuscript: 'data' is plural; the phrase 'is dependent upon' could be replaced with 'depends on' or 'requires.'

Paragraph one, subheading “Loss Of p38 MAPK Signaling Occludes Any Additional Effects Of Hypoxia On GLR-1 Localization”: the authors could dispense with the word 'dramatic'. They saw small differences that were not statistically significant and could simply say that.

'To test the second expectation…' could be changed to 'To test the second prediction'.

The authors could tighten up the last sentence of the Discussion, which is a bit repetitive. Suggestion: ‘Our findings highlight the idea that changes in kinase signaling could explain aspects of age-associated physiological decline’.

---

## [Author Response]

1) To highlight and contrast this pathway from other published views on oxygen sensing, please consider the following studies. There is some evidence for a role of guanylate cyclase homologue in oxygen sensation in C. elegans (Gray et al., Nature 430: 317-322, 2004). Also, there is a recent paper in Science Signaling elaborates a gas messenger-dependent acute oxygen sensing involving carbon monoxide-PKG-mediated phosphorylation of target enzyme as a critical step (Yuan et al., Sci. Signal 8(373): ra37, 2015). A pertinent discussion on these findings in comparison to the current work will further highlight the prevalence of context specific variations in oxygen sensing mechanisms.

We have added a paragraph to the Discussion section elaborating the difference between the oxygen-sensing mechanism described here and that of the soluble guanylate cyclase mechanism used in the aerotaxis response (i.e., the Gray paper). We have also added a paragraph to the Discussion section about the O2-CO-H2S sensory system employed by the carotid body (i.e., the Yuan paper). Citations to both publications are included.

2) The Discussion section needs some rearrangement. The importance of the identified O2-ROS-p38 MAP kinase phosphorylation (as shown in Figure 10) of the LIN-1/EGL-9 is lost amongst lengthy discussion on studies showing increased ROS formation under hypoxia by other investigators. Putative role of ROS under hypoxia in the current model of the study is only speculative and therefore can be shortened considerably.

We have removed the paragraph that describes the studies of other investigators regarding hypoxia and ROS formation. The sentences that describe two potential (but not proved) explanations for the role of PMK-1 in this pathway have been reorganized into a single paragraph.

3) Two key questions are posed in Figure 3. First, do PMK-1 and EGL-9 act in a common pathway? Second, is PMK-1 required for the effects of hypoxia on receptor localization? All the data needed to ask these questions are in panels E and F but the comparisons are not. The authors should indicate whether the difference between pmk-1 mutants, egl-9 mutants and pmk-1; egl-9 double mutants can be attributed to chance or not. Ditto for the differences between pmk-1 mutants raised in normoxic and hypoxic. To put it more simply: the asterisks in panels E and F don't help to make the comparisons.

We have used ANOVA with Tukey’s multiple comparison test to examine whether *egl-9* mutations enhance the phenotype caused by *pmk-1* mutations. We find that the number of GLR-1 aggregates in *pmk-1 egl-9* double mutants is not statistically greater than the number observed in *pmk-1* single mutants. We find that the number of GLR-1 synaptic puncta in *pmk-1 egl-9* double mutants is not statistically distinguishable from the number observed in *pmk-1* single mutants. We also used the same test to examine whether hypoxia can enhance the phenotype caused by *pmk-1* mutations. We find that the number of GLR-1 aggregates in hypoxic *pmk-1* animals is not statistically distinguishable from the number observed in normoxic *pmk-1* animals. We do find that the number of GLR-1 puncta on hypoxic *pmk-1* animals is statistically different from the number observed in normoxic *pmk-1* animals; however, the magnitude of that difference is small (a 19% decrease). This could indicate that hypoxia can alter GLR-1 puncta number by an additional mechanism other than through PMK-1 regulation; however, such an additional mechanism would be minor in comparison to the PMK-1-mediated mechanism. We have added these comparisons to Figure 3.

4) Please add some clarifications to the experiments that measure glutamate receptors in endosomes using co-localization with SYX-7 (Figure 2). First, these experiments determine the distribution of glutamate receptors in compartments in the cell body unlike every other experiment in which receptor distribution is determined in neurites. Is SYX-7 marking endosomes in neurites? If not, then the SYX-7-positive endosomes in the soma differ from the endosomes in neurites and what happens in these endosomes doesn't necessarily inform us about what happens in neurites.

Visualizing endosomes in *C. elegans* neurites using genetically encoded fluorescent proteins has remained a challenge in the field. The number of smooth endosomal compartments detectable in the interneuron neurites is relatively small (Rolls et al., 2002). Approaches using the GFP-tagged membrane residents of endosomes have typically be confounded by such molecules also decorating the neurite plasma membrane at high enough of a level as to obscure endosomal detection in underlying neurite (Hoerndli et al., 2013). The cell body is large enough to provide structural distance between the plasma membrane and endosomes, allowing us to visualize endosomal structures. This approach has been adopted in multiple previous publications (Zhang et al., 2012, Chun et al., 2008, Monteiro et al., 2012, Kowalski et al., 2011). Formally our results do not directly demonstrate that GLR-1 is accumulating at SYX-7-decorated endosomes in the neurites of *pmk-1* mutants, although the ability to suppress GLR-1 accumulation in *pmk-1* mutant neurites by blocking endocytosis is functional evidence that lends itself to this conclusion. Nevertheless, we have altered the text in the Results section concerning this experiment to reflect this uncertainty.

Also, the Methods should make clear how this colocalization was measured. It seems that the data are single optical sections through cell bodies neurons expressing both markers, but this isn't clear in the manuscript. Because the distribution of fluorescence within the cell is complicated and the colocalization between signals is not obvious it is critical that the reader understand how the data are analyzed to reach the conclusion that PMK-1 affects localization to endosomes.

We have clarified the Results section concerning colocalization and added an extensive description to the Materials and methods section of how the colocalization results were obtained.

5) In your model (Figure 10), please indicate the phosphorylation state of LIN-1 or EGL-9 under normoxia and hypoxia.

We have created a new figure that focuses on the pathway’s role in regulating GLR-1 recycling with the goal of making the figure simpler and easier to follow. In addition, as requested by the reviewers, we have indicated the hypothesized phosphorylation states of LIN-10 and EGL-9 in the model.

6) One of the reviewers had comments that suggested additional experiments, "low-hanging fruits" that would add considerably to the paper. We are not requiring these for acceptance, but ask that you consider whether you can do some or all of these rapidly, and if so, we would welcome their addition. But, in any case please address these comments.

We have completed the most straightforward experiments requested by the reviewer. A more extensive exploration of these questions would go beyond the scope of the current manuscript. Our conclusions from these experiments follow below.

*a) An interesting question is where PMK-1 is localized in these neurons. Is there a pool of p38 MAPK in the neurite that acts locally? The presented model suggests that this is what the authors are thinking, and this idea is readily testable.*

We have examined this by generating a transgene that expresses a PMK-1::GFP chimeric protein under the control of the *glr-1* promoter (thus restricting its expression to the command interneurons in question). We find that PMK-1::GFP is enriched in the cell body nucleus, but is also clearly found in the cytosol of the cell body and the ventral cord dendrites. The simplest model is that the abundant pool of PMK-1 in the dendrites can phosphorylate EGL-9 and other relevant targets in those same dendrites. However, this experiment cannot exclude the possibility that nuclear PMK-1 has a role with respect to regulating glutamate receptor trafficking. We have added this data to Figure 3.

*b) The authors propose that the p38 MAPK PMK-1 acts in a novel pathway to regulate glutamate receptor localization. This claim is based on the observation that the known upstream MAPKKK for PMK-1, NSY-1, and a known target of PMK-1 signaling, SKN-1, are not required for receptor localization. This is an interesting point and would benefit from additional data. The authors could test other genes linked to PMK-1 signaling, such as TIR-1 and ATF-7. Also, there are a small number of MAPKKKs encoded by the C. elegans genome. If the canonical MAPKKK is not regulating PMK-1 in this context perhaps one of the others is. There are a small number of predicted MAPKKKs in C. elegans. Do any MAPKKK mutants have defects in glutamate receptor localization?*

We have examined several different alleles of *tir-1* but not observed a GLR-1 localization defect. We have also examined mutants for the MAPKKKK *mlk-1* and *kin-18*, but we have not observed a GLR-1 localization phenotype. Similarly, mutations in *atf-7* do not affect GLR-1 localization, nor do they suppress the defects observed in *pmk-1* mutants.

We previously showed that mutations in the MAPKKK *dlk-1* can suppress the accumulation of GLR-1 observed in *lin-10* mutants (Park et al., 2009). However, this is the opposite phenotype one would expect for the loss of function of the MAPKKK that regulates PMK-1. Indeed, we showed in the same publication that DLK-1 acts as the MAPKKK for the MAPKK MKK-4 and the p38 MAPK PMK-3. This pathway acts on GLR-1 glutamate receptor trafficking by promoting GLR-1 endocytosis: mutations in the pathway result in depressed GLR-1 endocytosis, which suppresses the accumulation of GLR-1 observed in *lin-10* mutants, which are blocked for recycling.

Recently a role in the nervous system for the third p38 MAPK in *C. elegans, pmk-2*, was reported (Pagano et al., 2015). We examined *pmk-2* mutants for GLR-1 localization; however, we did not observe a phenotype. Given that PMK-1 and PMK-2 could act with partial redundancy, we also examined double mutants; however, *pmk-1 pmk-2* double mutants had a similar GLR-1 localization defect to that observed in *pmk-1* single mutants.

All of the data described above has been added as panels for the Figure 1 extended figure.

[Editors' note: further revisions were requested prior to acceptance, as described below.]

“… We generated a transgene containing the glr-1 promoter driving a full length PMK-1::GFP chimeric protein”. 'glr-1' needs italics.

We have made this change.

Throughout the manuscript: 'data' is plural;

The word “data” occurs twice in the manuscript: Paragraph one, subheading “CDK-5 Acts Downstream of p38 MAPK Signaling To Regulate GLR-1 Localization” and subheading “The p38 MAPK Pathway Regulates EGL-9 Subcellular Localization”. The word “data” used in the first instance is part of a phrase that is separate from the main subject and predicate of the sentence (i.e., “we observed”); thus, singular vs. plural verb agreement should not be an issue here. We have changed the verb “indicates” to “indicate” in the latter sentence so as to match the usage of the plural form of “data” on that page.

The phrase 'is dependent upon' could be replaced with 'depends on' or 'requires.'

The phrase “is dependent upon” is not present in the manuscript. We did notice “were dependent on” in subheading “The p38 MAPK Pathway Regulates EGL-9 Subcellular Localization”. We have kept “were” given the conditional nature and mood being set by this sentence. However, we have changed “dependent on” to “to depend on.”

Paragraph one, subheading “Loss Of p38 MAPK Signaling Occludes Any Additional Effects Of Hypoxia On GLR-1 Localization”: the authors could dispense with the word 'dramatic'. They saw small differences that were not statistically significant and could simply say that.

We have replaced “dramatic” with “statistically significant.”

'To test the second expectation…' could be changed to 'To test the second prediction'.

We have replaced “expectation” with “prediction.”

The authors could tighten up the last sentence of the Discussion, which is a bit repetitive. Suggestion: ‘Our findings highlight the idea that changes in kinase signaling could explain aspects of age-associated physiological decline’.

We have replaced the last sentence with the sentence suggested by the reviewers.